# Evaluation of Tunisian wheat endophytes as plant growth promoting bacteria and biological control agents against *Fusarium culmorum*

**Mouadh Saadaoui[1,2,3], Mohamed Faize[4], Aicha Rifai[4], Koussa Tayeb[4], Noura Omri Ben Youssef[3,5], Mohamed Kharrat[3], Patricia Roeckel-Drevet[1], Hatem Chaar[3,5], Jean-Stéphane Venisse[1] ***

**1** Université Clermont Auvergne, INRAE, PIAF, Clermont-Ferrand, France, **2** Université de Tunis El Manar, Campus Universitaire Farhat Hached, Tunis, Tunisia, **3** Field Crops Laboratory, National Institute for Agricultural Research of Tunisia, Tunisia, Tunisia, **4** Laboratory of Plant Biotechnology, Ecology and Ecosystem Valorization CNRST-URL10, Faculty of Sciences, University Chouaib Doukkali, El Jadida, Morocco, **5** National Institute of Agronomy of Tunisia, Tunis, Tunisia

\* j-stephane.venisse@uca.fr

## Abstract

Plant growth-promoting rhizobacteria (PGPR) applications have emerged as an ideal substitute for synthetic chemicals by their ability to improve plant nutrition and resistance against pathogens. In this study, we isolated fourteen root endophytes from healthy wheat roots cultivated in Tunisia. The isolates were identified based from their *16S rRNA* gene sequences. They belonged to *Bacillota* and *Pseudomonadota* taxa. Fourteen strains were tested for their growth-promoting and defense-eliciting potentials on durum wheat under greenhouse conditions, and for their *in vitro* biocontrol power against *Fusarium culmorum*, an ascomycete responsible for seedling blight, foot and root rot, and head blight diseases of wheat. We found that all the strains improved shoot and/or root biomass accumulation, with *Bacillus mojavensis*, *Paenibacillus peoriae* and *Variovorax paradoxus* showing the strongest promoting effects. These physiological effects were correlated with the plant growth-promoting traits of the bacterial endophytes, which produced indole-related compounds, ammonia, and hydrogen cyanide (HCN), and solubilized phosphate and zinc. Likewise, plant defense accumulations were modulated lastingly and systematically in roots and leaves by all the strains. Testing *in vitro* antagonism against *F. culmorum* revealed an inhibition activity exceeding 40% for five strains: *Bacillus cereus*, *Paenibacillus peoriae*, *Paenibacillus polymyxa*, *Pantoae agglomerans*, and *Pseudomonas aeruginosa*. These strains exhibited significant inhibitory effects on *F. culmorum* mycelia growth, sporulation, and/or macroconidia germination. *P. peoriae* performed best, with total inhibition of sporulation and macroconidia germination. These finding highlight the effectiveness of root bacterial endophytes in promoting plant growth and resistance, and in controlling phytopathogens such as *F. culmorum*. This is the first report identifying 14 bacterial candidates as potential agents for the control of *F. culmorum*, of which *Paenibacillus peoriae* and/or its intracellular metabolites have potential for development as biopesticides.

**Data Availability Statement:** All relevant data are within the paper and its Supporting Information files.

**Funding:** This work was awarded by the I-SITE CAP 20-25 (ANR grant 16-IDEX-0001) Emergence 2017 from the University of Clermont-Auvergne, and the project "Pack Ambition International 2019" [ANR grant n° P010O003] co-financed by the University of Clermont-Auvergne and the French Region "Auvergne-Rhônes-Alpes". The funders had no role in study design, data collection and analysis, decision to publish, or preparation of the manuscript.

**Competing interests:** The authors have declared that no competing interests exist.

## Introduction

Plants in nature undergo a wide range of biotic and abiotic stresses. To resist these threats, they have developed strategies to form alliances with beneficial plant-associated microbes through a signaling network mediated by phytohormones and a plethora of secondary metabolites [1, 2]. Beneficial microorganisms (fungi or bacteria) can be epiphytic, colonizing the surface of plant organs, or endophytic, living in the intercellular spaces and feeding on apoplastic nutrients without causing any apparent disease symptoms [3, 4].

The relationship between endophytes and their host may be neutral or even pathogenic (latent pathogens), but a subset of them has been found to develop beneficial behavior, conferring several physiological advantages on their host, and prompting better plant resilience. The positive effect of endophytes can be explained by several direct and indirect mechanisms. They include the production of plant growth regulators or phytohormones (auxins, cytokinins and gibberellins), improved biological nitrogen fixation and nutrient mobilization and assimilation (phosphorus, zinc, etc.), and the potentiation of tolerance to abiotic stress (salinity, drought, pollution), and biotic stress by stimulating the host to deploy specific defensive weapons locally and systemically. Endophytes may compete with pathogens either by physically interacting with them to become their prey (parasitic activities), or indirectly by colonizing infection sites and appropriating common nutrients, including the synthesis of substances with bactericidal action (HCN, "antibiotic" compounds, siderophores, etc.) or by inducing the accumulation of plant defenses (innate immunity) [5–7]. These capacities derive from the delivery or regulation of various complex and interconnected hormone signaling-dependent pathways, involved in growth (e.g., IAA, abscisic acid, gibberellic acid, and cytokinins) and plant defenses (e.g., salicylates, jasmonates, and ethylene) [8].

Endophytes are ubiquitous and have been associated with almost all plants examined [9]. Beneficial endophytes are mostly found in the rhizosphere, but they also colonize the phyllosphere, and a few are obligate symbionts transmitted to seeds [10]. The endophytic communities and the networks that drive them are thus remarkably diverse and complex. Factors affecting the composition of species inside microbial communities are far from fully understood [11]. However, beneficial endophytes associated with cereals have drawn interest because of their positive impact on crop growth and yield [12]. Their use is particularly relevant -if not urgent- for wheat (*Triticum aestivum* L.), one of the most important staple crops in the world, which is directly and profoundly affected by changing environmental conditions. The need to increase wheat production and the ecological damage caused by the excessive use of agrochemicals make endophytes an attractive alternative for sustainable production with low environmental impacts.

In Tunisia, durum wheat is a strategic crop because of its importance in traditional diet [13]. The cultivation of wheat started with Florence Aurore, a non-bearded soft wheat variety, which covered up to 80% of the Tunisian bread wheat surface until 1952. It has since been replaced by modern higher yield varieties [14, 15]. Unfortunately, varietal selection has adversaly affected the ability of these modern varieties to interact with native microbes, limiting the beneficial input from plant-microbe interactions, as reported by Valente et al. [16].

*Fusarium* species are the most destructive wheat pathogens and toxin-producing fungi, causing extensive damage and significant yield losses [17]. In Tunisia, *Fusarium culmorum* is one of the dominant wheat pathogens [18]. It is the causal agent of seedling, foot and root rot, and head blight (FSB, FFRR and FHB) on various small grain cereals, in particular wheat and barley [19]. The ability of *F. culmorum* to persist in the soil as chlamydospores for many years, the lack of fully resistant wheat cultivars and the fact that Tunisia is considered a climate change hotspot make this fungal agent difficult to manage [20, 21]. Fungicides are the most

effective tools to control diseases caused by *F. culmorum*. However, given their negative impacts on the environment and humain health, and the increased resistance of pathogens to tem, it is crucially urgent to find new safe alternatives [22].

Increasing agricultural productivity, particularly wheat production, in a sustainable and environment-friendly way is a major challenge. The use of PGPR, thanks to their ability to improve plant growth, productivity, and innate immunity, should help accomplish this aim. However, as awareness of environmentally friendly farming practices grows through the strict adoption of precautionary procedures with alternative biological controls, it is essential to find region-specific microbial strains that can be used as growth-promoting inocula to achieve desired crop production [23]. This means that specific selection of PGPR strains is necessary, according to the agronomic objectives pursued and the geographical areas assigned for their use. In this regard, the characterization of native PGPRs from agroclimatic areas where severe *F. culmorum* infestation occurs, and their valorization as biofertilizers and biopesticides against this fungal agent in its originating area and surroundings, are of great interest. This research focuses on bacterial endophytes colonizing the roots of an old bread wheat variety (Florence Aurore) cultivated in an organic farming system in Tunisia. The study had three main objectives: (i) verify the presence of beneficial bacterial root endophytes in this wheat variety through isolation and molecular identification, (ii) evaluate their ability to promote plant growth in a controlled chamber, measuring their capacity to produce indole-related compounds, HCN, and ammonia, and to solubilize phosphate and zinc elements, and (iii) characterize their ability *in vitro* to elicit plant defenses and act as a direct biocontrol agent against *F. culmorum*, particularly by inhibiting sporulation and macroconidia germination, given the importance of spores in the *Fusarium* life cycle and disease dissemination.

## Materials and methods

### Study site and sample collection

The study site was located in Gosset El Bey Bizerte, about 60 km north-west of Tunis (36°92'40.96"N, 9°69'90.89"E, 133 m a.s.l.). The topography is mainly hilly with stretches of plains. The climate is warm Mediterranean with dry summers. The annual temperature range is approx. 18.2°C, the average rainfall approx. 582.7 mm. Field sampling was conducted in May 2020. Healthy plant samples of wheat (*Triticum aestivum* L.) variety Florence Aurore were collected randomly from an organic crop field at grain-filling stage (Growth stage 87, according to the code defined by Zadoks et al. [24]. Samples were placed individually in plastic bags and brought to the Tunisian National Institute of Agronomic Research (INRAT) for bacteria isolation.

### Isolation of root endophytic bacteria

Plants were carefully removed from rhizospheric soil and washed with tap water. Roots were cut into sections 2–3 cm long and dried on filter paper. All root segments were surface-sterilized to remove epiphytic microbes by dipping them in 70% ethanol for 2 min, in 0.5% NaOCl for 2 min, in 70% ethanol for 1 min, and then washing them three times with sterile distilled water. Fragments were placed in tryptone soya agar (TSA). Bacteria were also isolated by serial dilution plating of sterilized crushed root samples on TSA plates as described by Hameed et al. [25]. Plates were incubated at 28±2°C and inspected daily until bacteria appeared. The surface sterilization technique was checked by inoculating TSA plates with 1 mL of the wash water. Absence of microbial growth was taken as evidence of successful surface sterilization. Isolates were picked and transferred onto agar-solidified lysogeny broth (LB) medium for further purification. The morphological aspect of the 14 strains used in this work is shown in S1 Fig.

**DNA extraction, *16S rRNA* gene sequencing, and phylogenetic analysis.** Total genomic DNA was extracted from bacteria using the phenol-chloroform method. A partial region of *16S rRNA* gene was amplified using 27/1492R primers [26]. PCR amplifications were performed using a BioRad thermal cycler with initial denaturing at 95˚C (2 min), followed by denaturing for 35 cycles of 95˚C for 30 s, primer annealing at 58˚C for 30 s, primer extension at 72˚C for 45 s, and a final extension at 72˚C for 5 min. The PCR reactions were run in a total volume set at 50 µL, the mix containing 1 µL of genomic DNA (25 ng µL$^{-1}$), 10 µL of colorless GoTaq Felxi Buffer (5X), 3 µL of $MgCl_2$ solution (25 mM), 1 µL of PCR nucleotide mix (10 mM each dNTP), 1 µL of primers (5 µM each), 0.25 µL of GoTaq (G2 Flexi DNA Polymerase 5 u µL$^{-1}$), and 32.75 µL of sterile water. PCR products were analyzed by agarose gel electrophoresis (1.5%) and sequenced by Genewiz company (Germany). *16S rRNA* gene sequences of each isolate were identified by BLAST alignment inferred in the public database NCBI (*https://blast.ncbi.nlm.nih.gov/Blast.cgi*), and recorded in the NCBI GenBank database. Related accession numbers are listed in Fig 1A. Homologous *16S rRNA* gene sequences used in the study were retrieved from the NCBI BLAST analyses. Multiple DNA sequence alignments were performed using the Clustal-W algorithm. Maximum likelihood analysis was used to construct a phylogenetic tree as implemented in the PhyML program, with bootstrap analysis based on 1000 random resamplings. Nucleotide sequence accession numbers are listed in Fig 1A.

## Extracellular enzyme production

**Catalase.** A small quantity of fresh bacterial culture was taken from an LB plate using a sterilized wooden pin and placed on a glass slide. A drop of hydrogen peroxide (3%) was added to the samples. Immediate bubbling showed catalase production [27].

**Amylase.** Overnight-grown bacterial cultures were used for spot inoculation on mineral starch medium agar containing (g L$^{-1}$): $K_2HPO_4$ 0.5, $KH_2PO_4$ 1, $NH_4Cl$ 1, $MgSO_4$ 0.2, starch 5, agar 20 (pH 8.0) [28]. For each strain, four spots per plate were run (each spot 10 µL). After 48 h of incubation at 28˚C, the plates were flooded with an aqueous solution of iodine 1% (w/v) in potassium iodide 2% (w/v) to reveal the clear zone of degradation around colonies producing amylase.

**Protease.** LBA medium supplemented with 1% skim milk was used to detect protease activity [29]. Fresh bacterial cultures were spot-inoculated and plates were incubated at 28˚C for 3–4 days. The presence of halo zones around the bacterial growth was considered as a positive result for protease production.

**Cellulase.** Cellulase activity was tested by spot inoculation on M9 minimal salt medium comprising (g L$^{-1}$) $Na_2HPO_4$ 33.9, $KH_2PO_4$ 15, NaCl 2.5, $NH_4Cl$ 5, yeast extract 1.2, and agar 15 amended with 1% carboxymethyl cellulose [30]. Plates were left at 28˚C for 7 days, then flooded with an aqueous solution of iodine 1% (w/v) in potassium iodide 2% (w/v). The presence of a clear zone around the colonies was considered as a positive result for cellulase production.

**Pectinase.** The pectinase activity was assessed through spot inoculation on M9 minimal salt medium supplemented with 1% (w/v) pectin [30]. Following seven days of incubation at 28˚C, colonies were treated with a 0.12% Congo red solution. The presence of a clear zone surrounding the colonies was indicative of positive pectinase-producing isolates.

## *In vitro* determination of plant growth promotion activities

**Production of indole-related compounds.** Overnight-grown cultures of the bacterial isolates were inoculated in LB broth supplemented with 0.1% of *L*-tryptophan. After 48 h of growth at 28˚C under constant shaking (240 rpm), 1 mL of supernatant was collected from

**(a)**

| Phylum | Isolated Species | Accession number (NCBI) | Best hit - Related strain | % identity | CDS length |
|---|---|---|---|---|---|
| *Pseudomonadota* | *Achromobacter* sp. | OP595769 | *Achromobacter* sp. MYb9 (KU902435.1) | 99.64% | 1425 |
| *Bacillota* | *Bacillus cereus* | OP595775 | *Bacillus cereus* LXJ76 (MN746191.1) | 99.65% | 1460 |
| | *Bacillus halotolerans* | OP595771 | *Bacillus halotolerans* strain ROA207 (MZ317478.1) | 99.57% | 1467 |
| | *Bacillus megaterium* | OP595774 | *Bacillus megaterium* strain SCSIO_43708 (MH283792.1) | 99.59% | 1466 |
| | *Bacillus mojavensis* | OP595778 | *Bacillus mojavensis* strain A21 (EU366229.1) | 99.79% | 1467 |
| | *Bacillus subtilis* | OP595777 | *Bacillus subtilis* strain TBS 1 (MT197332.1) | 94.51% | 1425 |
| *Pseudomonadota* | *Enterobacter hormaechei* | OP595783 | *Enterobacter hormaechei* strain L51 (CP033102.1) | 99.49% | 1392 |
| *Bacillota* | *Paenibacillus peoriae* | OP595788 | *Paenibacillus peoriae* strain D71_SO3R (MK883174.1) | 99.78% | 1389 |
| | *Paenibacillus polymyxa* | OP595789 | *Paenibacillus polymyxa* strain R191 (MK100924.1) | 99.01% | 1441 |
| *Pseudomonadota* | *Pantoea agglomerans* | OP595791 | *P. agglomerans* strain LA131 (MN006238.1) | 85.35% | 1194 |
| *Pseudomonadota* | *Pseudomonas aeruginosa* | OP595793 | *Pseudomonas aeruginosa* strain HL2 (KF413420.1) | 100.00% | 1350 |
| | *Pseudomonas frederiksbergensis* | OP595794 | *Pseudomonas frederiksbergensis* (MN865449.1) | 99.65% | 1450 |
| *Pseudomonadota* | *Rhizobium* sp. | OP595796 | *Rhizobium* sp. Strain IAUK3223 (MK212373.1) | 92.30% | 1295 |
| *Pseudomonadota* | *Variovorax paradoxus* | OP595801 | *Variovorax paradoxus* strain JZY4-58 16S (MT102304.1) | 99.07% | 1399 |

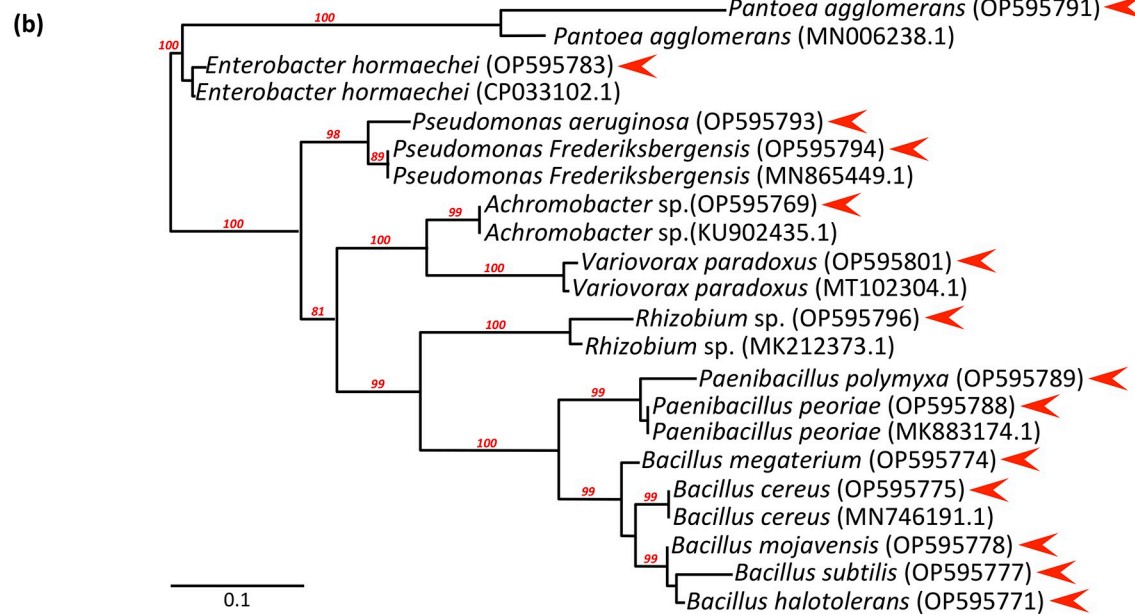

**Fig 1. Molecular characterization of the isolated root endophytic bacterial strains selected for this work.** (**A**) Detailed molecular characteristics of the isolated root endophytic bacterial strains. (**B**) *16S RNA*-based molecular phylogenetic distribution of the bacterial isolates (red triangle) with phylum-closed sequences retrieved from NCBI was generated using maximum likelihood analysis implemented in the PhyML program. Values of the bootstrap analysis (1000 repetitions) are given at the nodes. Bar 0.1 corresponds to the nucleotide substitution per sequence position. Morphological appearance of the new strains is shown in S1 Fig.

each culture after 5 min of centrifugation at 14,000 rpm, and mixed with 2 mL of Salkowski's reagent (FeCl₃ 0.5 M, and 35% HClO₄). Samples were incubated at room temperature for 30 min and the intensity of color was measured by spectrophotometry at 535 nm. This assay was carried out in triplicate. Non-inoculated LB broth (with *L*-tryptophan) was used as control. A standard curve with pure IAA was generated to estimate the amount of produced indole-related compounds, produced by suspending IAA (Sigma) in 100% ethanol at a concentration of 10 mg mL$^{-1}$ and diluting it in LB medium to a concentration of 10 – 100 μg mL$^{-1}$. Control was sterile LB medium with and without *L*-tryptophan [31, 32].

**Phosphate solubilization.** Inorganic phosphate solubilization ability was detected by spot inoculating isolates on Pikovskaya's agar medium (PVK) amended with 0.5% (w/v) Ca₃(PO₄)₂, consisting of (g L$^{-1}$) glucose 10, yeast extract 0.5, (NH₄)₂SO₄ 0.5, MgSO₄.7H₂O 0.1, KCl 0.2, and agar 15 (pH 7) [33]. Plates were incubated at 28°C for 5–7 days. The formation of a clear zone around the colonies indicated the solubilization of phosphate by the isolates.

**Ammonia production.** Ammonia was estimated by growing bacterial strains in peptone water for 72 h at 28°C. Cultures were centrifuged for 5 min at 14,000 rpm, and to 1 mL of each culture supernatant 50 μL of Nessler's reagent was added (KI 7%, HgI2 10%, 50% aqueous solution of NaOH 32%) [34]. The formation of a slight yellow to brownish color is an indicator of ammonia production. The optical density of each strain was measured at 450 nm to quantify the amount of ammonia produced using a standard curve of $(NH_4)_2SO_4$ solution at the following concentrations: 0.1, 0.5, 1, 1.5, 2, 2.5, 3, 3.5, and 4 μmol mL$^{-1}$.

**Zinc solubilization.** Bacterial strains were screened for their zinc solubilization potential on Tris-minimal medium containing (g L$^{-1}$) *D*-glucose 10, Tris-HCl 6.06, NaCl 4.68, KCl 1.49; NH$_4$Cl 1.07, Na$_2$SO$_4$ 0.43, MgCl$_2$.2H$_2$O 0.2, CaCl$_2$.2H$_2$O 0.03, agar 15 (pH 7), amended with 0.1% zinc oxide according to Fasim et al. [35]. Plates were incubated for 14 days in the dark at 28°C. Zinc-solubilizing strains produced halo zones around colonies. The surface of these zones was measured using ImageJ software to calculate the Zn solubilization efficiency (SE) % according to Sharma et al. [36] using the following modified equation: Zn solubilization efficiency SE (%) = (surface of solubilization halo / surface of colony) x 100.

**Production of hydrogen cyanide.** HCN production was assessed by cultivating bacterial colonies on LB medium supplemented with 4.4 g L$^{-1}$ of *L*-glycine [37]. Sterilized filter papers were soaked in alkaline picrate solution (0.25% picric acid and 1.25% sodium carbonate) for one minute, placed on the cover of Petri plates, and sealed with Parafilm. After 4–5 days of incubation at 28°C, color change of the filter paper from yellow to brick red indicates the production of HCN. This assay was repeated four times.

## *In vitro* biocontrol activity

**In vitro dual-culture analysis.** Antifungal activity of bacterial isolates against *Fusarium culmorum* was assessed by dual culture assays as described by Al-Mutar et al. [38]. Four sterile filter paper discs (diameter 5 mm) were soaked on overnight-grown bacterial culture and placed symmetrically around a 5 mm diameter agar plug of *F. culmorum* (7 days old) in the center of potato dextrose agar (PDA) medium. PDA dishes inoculated only with *F. culmorum* served as control. Plates were incubated for 5–7 days at 28°C. Growth radius of *F. culmorum* towards bacterial colonies was measured daily until no further growth was detected. The inhibition rate (IR) was calculated by IR = R1-R2 / R1 x 100, where R1 is the colony radius of *F. culmorum* in the control plate and R2 is the colony radius of *F. culmorum* in the presence of bacterial strains. For each bacterial isolate, five replicates were run and the whole experiment was repeated twice.

**Antibiosis of selected bacterial strains.** The five isolates selected based on their growth inhibition percentage calculated during the dual culture assay were further studied for their *in vitro* antibiosis activity [39]. Liquid LB media (25 mL for each repetition) were inoculated with 1 mL of fresh bacterial culture (OD at 595 nm = 1) in 50 mL conical centrifuge tubes. After 48 h of incubation at 37°C under constant shaking, culture media were centrifuged at 8000 x g for 5 min to collect the supernatant then sterilized by three steps of filtration (0.45 μm once and 0.2 μm twice). PDA medium with 30 g of agar per liter was prepared then kept at 55°C in a water bath, 20 mL of each bacterial filtrate was mixed with the same volume of PDA medium then poured onto Petri plates. Three replicates for each treatment were run and for the control plates PDA medium, which were only mixed with a filtrated LB media. All plates were inoculated with mycelial plugs of *F. culmorum* and incubated for 7 days in the dark at 28°C. The growth surface was calculated using ImageJ software. The growth inhibition percentage was calculated using the formula: Inhibition percentage = Growth surface in the control plates – Growth surface in the treated medium / Mean of growth surface in the control plates x 100.

**Inhibition of macroconidia formation.** To assess the formation of *F. culmorum* macroconidia in the presence of the five selected bacterial strains, plates of the dual confrontation assay were kept at 28˚C for one month and agar blocks (4 mm in diameter) were excised from the *F. culmorum* growth surface, transferred to tubes containing 1 mL of sterilized water (two plugs per tube in three technical replicates) and vortexed for 30 s. The number of macroconidia was determined under a light microscope using KOVA Glasstic slides, as described by Semighini et al. [40].

**Effect of selected bacterial strains on *F. culmorum* macroconidia germination.** The effect of bacterial strains on the germination of *F. culmorum* macroconidia was evaluated with two methods, using the living bacteria or their intracellular metabolites.

For the experiment using living bacteria, we used the method described by Bruisson et al. [41] with some modifications. Briefly, *F. culmorum* spores were obtained from a PDA culture and suspended in a PDB medium. The spore concentration was then adjusted to $2.10^5$ spores $mL^{-1}$. A 100 μL aliquot of this sporal suspension was mixed with 100 μL of fresh bacterial culture in sterilized liquid LB medium (OD at 595 nm = 1). Control samples consisted of 100 μL of spore suspension ($2.10^5$ spores $mL^{-1}$) mixed with 100 μL of sterilized liquid LB medium. Each treatment was replicated three times. The resulting 200 μL mixture (total volume) was inoculated onto PDA medium and incubated at 28˚C for 5 days for visual observation of mycelial growth. To confirm macroconidia germination at a macroscopic level, the same mixture was placed in Eppendorf tubes under the same conditions. A drop of this mixture was then placed on a glass slide, and the germination examined under an optical microscope.

In the second approach, the inhibitory effect of the intracellular metabolites was assessed. Fresh bacterial cultures from LB liquid medium were centrifuged at 8000 x g for 15 min. The supernatant was removed and the pellet was retained. To obtain the intracellular fraction, the pellet containing the bacterial cells was washed with a 1% NaCl solution to remove any remaining supernatant. It was then resuspended in 800 μL of TEP buffer (10 mM Tris, pH 8.0, 1 mM EDTA and 0.001% (w/v) phenylmethylsulfonyl fluoride), and the final optical density (OD) at 595 nm was adjusted to 1. The pellet was sonicated for one minute with 3 cycles of 10 seconds pulse on/off on ice using a Vibra Cell™ VCX 130 ultrasonic processor with an amplitude of 50%. After sonication, the samples were centrifuged for 30 min at 12,000 rpm. The supernatant was filtered twice using a 0.2 μm filter, and the intracellular fraction was stored at 4˚C for further use.

For the assay, we used the method described by Bruisson et al. [41] with slight modifications: 4.95 μL of the intracellular fraction was mixed with 1.65 μL of *F. culmorum* macroconidia (suspended in PDB medium) at a final concentration of $2.10^5$ spores $mL^{-1}$ on KOVA Glasstic slides. The control was 4.95 μL of TEP buffer mixed with 1.65 μL of *F. culmorum* macroconidia (suspended in PDB medium) at a final concentration of $2.10^5$ spores $mL^{-1}$. This assay was performed five times and repeated twice. The KOVA Glasstic slides were placed in a humid box at 28˚C, and observed under an optical microscope at 6, 12, 24 and 96 hours. The analysis criteria were the percentage of germinated macroconidia (%$g$), germ tube length corresponding to hyphae length (Lh in μm) per square (0.33 mm x 0.33 mm) in KOVA Glasstic slides measured using ImageJ, and the morphology of hyphae [41].

## Bioassay and plant material

The effect of isolated bacteria on durum wheat was tested interm of seedling growth in a greenhouse for 3 weeks. The variety "Karim", one of the most widely used and productive durum wheat varieties in Tunisia, was chosen for this study. Karim seeds were provided by the National Institute of Agronomic Research of Tunisia (INRAT).

## Seeds sterilization and inoculum preparation for the *in vivo* and *in planta* assays

Fresh bacterial cultures were scraped off with 3 mL of sterile distilled water, placed in 10 mL of Luria Bertani (LB) broth, and shaken for 24 h. This medium was also used for the preparation of inocula suspensions to ensure the viability of bacteria after inoculation. The final concentration of the viable bacteria was adjusted to $10^8$ CFU mL$^{-1}$ just before inoculation. Wheat seeds were surface sterilized as described by Fernandez and Chen [42] with minor modifications. Seeds were washed with tap water, sterilized by immersion in 0.5% sodium hypochlorite for 5 min, and washed three times with sterile distilled water. After drying overnight on filter paper under aseptic conditions, sterilized seeds were placed in sterile dishes for further use.

**Evaluation of plant growth variables and ecophysiology trait estimation under greenhouse conditions.** Sterilized wheat seeds were soaked in 20 mL of bacterial suspension ($10^8$ CFU mL$^{-1}$) for 24 h with slow shaking [43], and sown in pots 6,5 cm x 6,5 cm (one seed per pot for six repetitions) filled with 1/3 sterilized sand and 2/3 soil. The pots were stored in a controlled greenhouse under the following conditions: 16/8 h light/dark photoperiod under an illumination of photosynthetic photon flux density 150 μmol m$^{-2}$ s$^{-1}$, a temperature of $23 \pm 0.5°C$, and a relative humidity of $65 \pm 5\%$. One week after seedling emergence, 1 mL of inoculum ($10^8$ CFU mL$^{-1}$) was applied at the base of each seedling. Seedlings inoculated with only sterile LB served as controls. All pots were placed in a greenhouse and watered on alternate days with tap water.

After 21 days under greenhouse conditions, plants were gently harvested, and roots were removed from the soil and washed with tap water. The morphological variables were recorded, including length and fresh weight of roots and shoots. Biomass was oven-dried at 60°C for 3 days, and dry weight was measured. At the same time, 0.5 g samples of root and leaf tissues were taken for the analysis of defense related genes. These samples were promptly treated with liquid nitrogen and frozen at -80°C for RNA extraction.

To quantify the content of chlorophylls (Chl) and epidermal flavonols (Flav) and their ratio, the nitrogen balance index (NBI), a non-destructive method was used, consisting of a Dualex sensor (Force-A, France) performing instantaneous quantification [44]. Measurements were made on uniform, fully developed light-exposed leaves of the same plants.

**Total RNA isolation, cDNA synthesis and quantitative PCR of defense-related genes.** To determine the expression levels of candidate genes inherent to both plant growth and immune performance, total RNA was extracted from roots and leaves of 15 plants used in the above experimental setup, randomly distributed beforehand into three biological replicates. Organs were carefully sampled, immediately frozen in liquid nitrogen, and stored at -80°C. Samples were ground to a fine powder with three metallic balls (diameter 5mm) for 90 s in Eppendorf tubes. Total RNA was extracted from 25 mg of samples and treated with a CTAB extraction buffer (cetyltrimethylammonium bromide) as previously described in Ben Amira et al. [45]. RNA concentration and purity were determined using a NanoDrop™ Spectrophotometer ND-1000 (NanoDrop®, France). First-strand cDNA was synthesized using 2 μg of purified total RNA with Oligo-dT using the SuperScript® III First-Strand Synthesis System for RT-PCR (Invitrogen, Carlsbad, CA, USA), and stored at -20°C for further use.

Real-time quantitative PCR was carried out on an Applied Biosystems StepOnePlus™ Real-Time PCR System (Applied Biosystems, Foster City, CA, USA). The amplification reactions were performed with 15 μL of final mixture consisting of 12.5 μL of Takyon™ Rox SYBR® MasteMix dTTP Blue (Eurogentec, Liege Science Park, Belgium), 1 μL of 1:30 diluted cDNA, and 100 nM primers (detailed in S1 Table). *q*RT-PCR reactions were set up with the following thermal cycles: pre-denaturing at 94°C for 5 min, and 40 cycles of 94°C for 10 s,

60˚C for 15 s, and 72˚C for 10 s. PCR reactions were ended by generating a dissociation curve of 60–95˚C with an increment of 0.3˚C / 15 s to ensure primer dimers and nonspecific amplifications. For each primer pair, PCR conditions were determined by comparing threshold values in a dilution series of the RT product (0, x10, x20, x40, x80), followed by non-template control for each primer pair. Specific amplification of only one desired band was observed using each primer combination for RT-*q*PCR analysis, and PCR efficiency was calculated for each primer pair. The amplification of five wheat encoding-housekeeping genes (Actin (*ACT*), glyceraldehyde-3-phosphate dehydrogenase (*GAPDH*), heterogeneous nuclear ribonucleoprotein (*RPN*), RNaseL inhibitor-like (*RLI*), and *26S* ribosomal RNA (*26S rRNA*)) were used as internal controls for normalizing all amplification data. These genes were chosen from a range of widely-used housekeeping genes, and specifically selected for their stable expressions during PGPR treatments. Each referrer belongs to protein families involved in different cell processes to minimize the risk of co-regulations. The $2^{-\Delta\Delta Ct}$ method was used to evaluate the relative expression of defense-related genes [46]. PCR reactions were performed in triplicate.

## Statistical analysis

Statistical analyses were performed using R software (version 4.2.1) [47], with the application of the stats packages rstatix and bestNormalize. Data normality and homogeneity of variance were examined with Levene's test and the Kolmogorov-Smirnov test, respectively, together by both visual inspection of the residuals (Q-Q plot: quantile-quantile plot plot), and significance test (Shapiro-Wilk's test). In most cases, normality of the residuals was respected. When normality and homoscedasticity were assumed, the data underwent a standard statistical method of analysis of variance (ANOVA), followed by Tukey's HSD post-hoc test ($p \leq 0.05$). When normality and homoscedasticity did not hold, the Kruskal-Wallis test based on ranks of the data was used. Epsilon squared was used as the measure of the Kruskal-Wallis test effect size [48]. It ranged between 0 and 1 and indicated the ratio of variance in the response variable explained by the explanatory variable, the Strain factor. Kruskal-Wallis test was followed by Dunn's post-hoc test. In *q*PCR analyses, $Log_2\Delta\Delta^{Ct}$ values were calculated for each sample. Relevant data transformations were performed, and homogeneity of variances was verified on transferred data. For leaves and roots, a multiple *t*-test was applied to determine significant differences ($p \leq 0.05$) between untreated and PGPR-inoculated organs, and *p*-values were adjusted using the Benjamini-Hochberg (BH) procedure.

## Results and discussion

### Diversity of the isolated bacterial endophytes

The plant growth-promoting potential of rhizobacteria isolated from wheat roots grown in an organic farming system in Tunisia and their *in vitro* biological control capacity against *F. culmorum* (the causal agent of fusarium seedling and head blight diseases, FSB and FHB, resp.) was examined and characterized in a multidisciplinary approach.

Fourteen endophytic bacterial strains of healthy wheat root were isolated using the enrichment culture technique, one of the conventional approaches using Tryptic soy agar medium, and then underwent molecular, physiological and biochemical characterizations. The culture-dependent method recovered representative genera that after subsequent testing of their functional abilities both *in vitro* and *in planta* could offer a promising strategy for the development of new beneficial inoculants. The macroscopic appearance of the isolates is shown in S1 Fig.

The molecular characterization of each bacterial strain was based on the sequencing of the *16S rRNA* gene. The subsequent Blast analysis from the NCBI gene databank (Fig 1A) combined with a global phylogenetic distribution using the maximum likelihood analysis

implemented in the PhyML program (Fig 1B), showed that strains fell into two phylum-divisions: *Pseudomonadota* and *Bacillota*. *Bacillota* was found to be the predominant endophytic bacterial phyla in our study. In these two phyla, eight genera were identifiable; namely *Achromobacter*, *Enterobacter*, *Paenibacillus*, *Pantoea*, *Pseudomonas*, *Rhizobium*, *Variovorax* and *Bacillus*. The last genus is the most abundantly represented with five different species. Species of the genus *Paenibacillus* were originally considered to belong to the genus *Bacillus*, based on morphological characteristics, but were later reclassified as a separate genus in 1993 [49, 50]. The predominance observed here echoes the finding that bacilli from *Bacillota* form the dominant bacterial class with Gammaproteobacteria of the *Pseudomonadota* in the endophyte community in aerial and underground parts of wheat [51].

In detail and at species grade level, NCBI best hits showed that the percentage identity of 11 strains out of 14 exceeded 99%, but was less than 95% for the rest of the strains. The phylogenetic analysis with phylum-closed sequences retrieved from NCBI shown that our isolated strains were probably *Achromobacter* sp., *Bacillus cereus*, *Bacillus halotolerans*, *Bacillus megaterium*, *Bacillus mojavensis*, *Bacillus subtilis*, *Enterobacter hormaechei*, *Paenibacillus peoriae*, *Paenibacillus polymyxa*, *Pantoea agglomerans*, *Pseudomonas aeruginosa*, *Pseudomonas frederiksbergensis*, *Rhizobium sp*., and *Variovorax paradoxus*.

The *16S rRNA* gene sequence is the most common housekeeping genetic marker and has been used to study bacterial phylogeny and taxonomy for several reasons. Firstly, this gene is present in nearly all bacteria, often occurring as a multigene family or operons. Also, the functional role of the *16S rRNA* gene has remained consistent over time, indicating that random sequence variations serve as a more accurate measure of evolutionary time. Lastly, the *16S rRNA* gene, spanning approximately 1,500 base pairs, offers a large enough dataset for computational analysis [52]. The usefulness of *16S rRNA* gene sequencing as a tool in microbial identification is well-established, although it may have limited phylogenetic power at the species level and poor discriminatory power for some genera, especially for strains that belong to common sub-species clades [53]. In addition, it is dependent on the deposition of complete unambiguous nucleotide sequences into databases and applying the correct "label" to each sequence [54]. Interestingly, some bacterial strains had a homology less than 97%, implying that they may be new species [55].

The "core microbiome" is a term now widely used to refer to members that are consistent features of a dataset hypotheisized to reflect underlying functional relationships of a specific bacterial population with the host [56]. Determining this core is challenging for lack of an appropriate method. Furthermore, we emphasive that the 14 species in our study are not exhaustive and the absence of some bacterial taxa does not mean that they are not present in our samples. However, the diversity and dominance of certain taxa (known as major taxa) observed in previous studies were also observed in our study. *Pseudomonas* was among dominant endophytic genera in *T. aestivum* in roots, leaves, and coleoptiles, while *Paenibacillus*, which is a core genus specific to *T. aestivum*, and *Variovorax* were subdominant bacterial genera hosted by these tissues [57]. In our study, bacilli showed the greatest diversity of species. This predominance may be due to the ability of *Bacillus* species to form endospores that survive under harsh conditions and to produce bactericidal substances that inhibit competitors in the rhizosphere [58].

The microorganisms that interact with plants are mainly common species and in some cases niche-specific. Some species that belong to the *Achromobacter*, *Pseudomonas*, *Pantoea*, and *Variovorax* genera are among the most typical wheat endophytes [59]. Species such as *B. subtilis*, *P. polymyxa* and *P. aeruginosa* have been reported in different parts of the plant including internal tissues [60, 61]. Comby et al. [51] observed that *B. megaterium* with *Microdochium bolleyi* and *Gaeumannomyces gramis* were indicator species characteristic of wheat

roots, while *B. mojavensis* was a specific species of the rhizosphere colonizing rhizoplane and spaces between cortical cells [62, 63]. Other genera such as *Enterobacter*, *Paenibacillus*, *Rhizobium*, and *Pseudomonas* are also related to the wheat rhizospheric microbiome [61, 64].

## Plant growth promoting abilities of the isolated root endophytes

We examined the plant growth-promoting abilities of the 14 isolated endophytic bacteria for their potential as bio-inoculants to wheat plants. To this end, bacterial strains were individually inoculated to wheat seeds before sowing under controlled conditions. After 21 days of growth in pots, results showed that all root-associated bacterial strains exerted significant positive effects on the growth and several ecophysiological variables. Compared with control plants treated with sterile LB, each strain showed at least one significant stimulatory effect out of the eight measured growth variables. Interestingly, all these species have been reported as plant growth-promoting agents for wheat and/or other plant species, such as fenugreek (*Achromobacter sp.*), lily (*B. halotolerans*), chickpea (*B. subtilis*), bean (*Enterobacter hormaechei*), grapevine (*Paenibacillus peoriae*), tomato (*Paenibacillus polymyxa*), pepper (*Pseudomonas Frederiksbergensis*), and alfalfa (*Variovorax paradoxus*) [65–72].

The most positively affected variables were shoot fresh weight, and root fresh and dry weight. Eleven strains out of the 14 strains tested significantly increased shoot and root biomass compared with the controls. The least affected variable was leaf epidermal flavonol content, in which no significant effects were observed for any of the inoculated plants (Fig 2B–2D). Flavonoids and related phenolic compounds are secondary metabolites produced by plants in response to various stresses and play a major role in improving plant-microbe interactions [73, 74]. Flavonoids are thought to regulate the endophytic microbial community in roots. A flavonoid accumulation might increase the environmental pressure on endophytes, thus affecting the endophytic community [75] because some flavonoids, known to be antibacterial agents, can inhibit the growth of many microorganisms [76]. Here we detected no significant increase or decrease in leaf epidermal flavonol content of any of the inoculated wheat seedlings compared with the controls, although some trends were apparent with *B. megaterium*, *P. peoriae*, *P. agglomerans*, and *Rhizobium* sp (Fig 2C). Given that falvonols, which are primary flavonoids in nature, play a key role in plant-endophyts interactions, we can assume that the symbiosis of these endophytic bacteria and wheat seedlings was not profoundly affected.

The increased root biomass could be related to the bacterial production of indole-related compounds (IRCs). All strains tested positive for the IRCs, with ranging production levels. The highest amount was recorded for *P. agglomerans* (37.40 μg mL$^{-1}$); this species was reported as a *T. aestivum* root-growth promoting agent [77]. Several factors could explain these variations of IRC production between strains, including the differential modulation of the biosynthetic pathways between strains and/or the thermodynamic features of enzymes that convert primary heterocyclic aromatic organic compounds (*L*-tryptophan) into related indole conjugate forms [78]. Every root-associated endophytic bacterium is able to produce IRCs, which potentially could include indole-3-acetic acid (IAA). IAA helps potentiate plant nutrition by absorbing nutrients from the soil through enhancing root development [79]. Consequently, this phytohormone is considered as the main effector substance in plant-bacteria interaction, phytostimulation, and plant immunity [80]. Furthermore, the *in vitro* production of IRCs in the presence of *L*-tryptophan, which is generally considered an IAA precursor [81], might not reflect the *in situ* processes, as plants and root endophytes interact as a holobiome, and/or other organisms from the endophytic community might participate. Additional analysis such as high-pressure liquid chromatography (HPLC) is needed to validate the hormonal factor influencing wheat root growth [82].

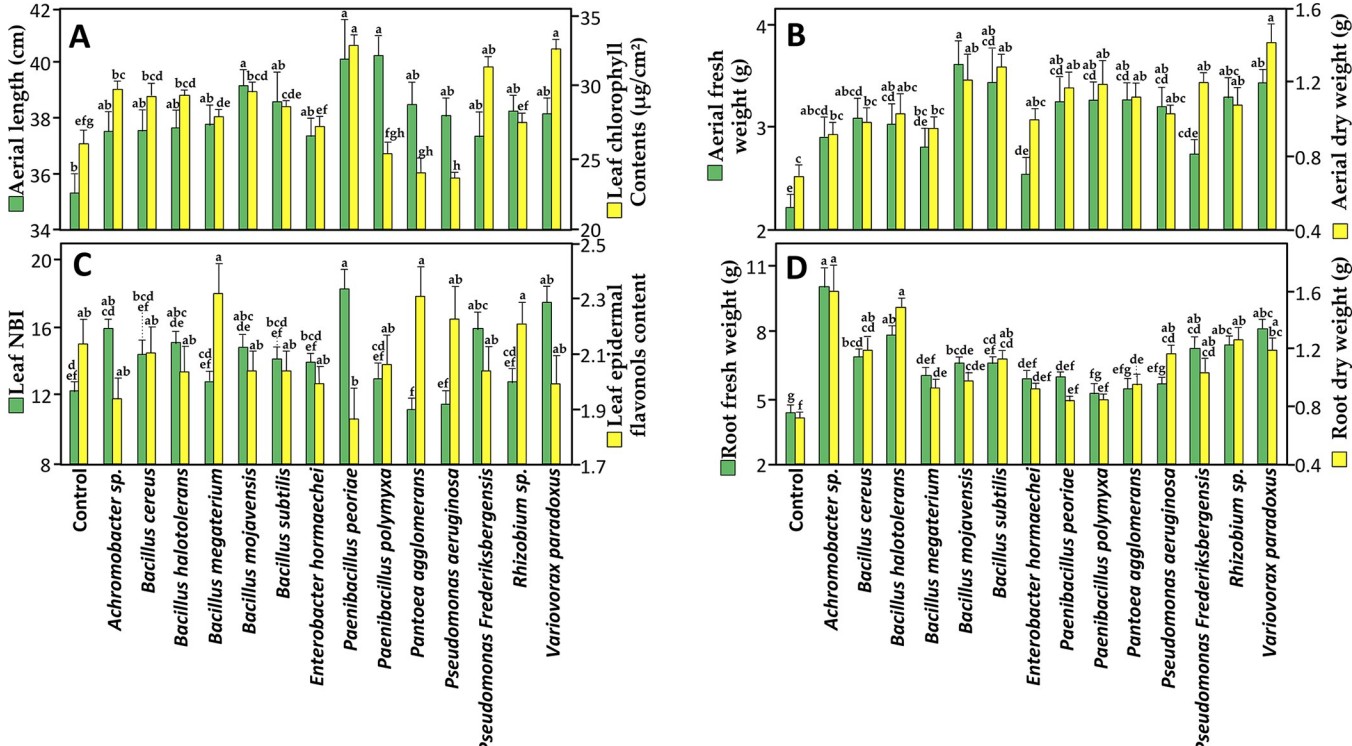

**Fig 2. Effect of root endophytic bacterial strains on wheat physio-morphological variables.** **(A)** Aerial length and leaf chlorophyll contents, **(B)** aerial fresh and dry weight, **(C)** leaf NBI and epidermal flavonol content, and **(D)** root fresh and dry weight of wheat plants (Karim variety) after 21 days of cultivation under greenhouse conditions. Data correspond to the mean of a minimum of ten repetitions ± SE. Different letters above data show significant differences within rows ($p \leq 0.05$).

Inoculation with 11 strains out of the 14 tested isolates gave significantly higher values of shoot fresh weight than non-inoculated plants, and to in a lesser extent of shoot dry weight and length. These data highlight the potential of these bacterial strains to enhance shoot development and biomass accumulation (Fig 2A and 2B). Interestingly, this physiological effect can be correlated with the leaf chlorophyll contents, one of the key regulators of efficient photosynthesis [83]. Wheat seedlings had significantly higher values than the controls when inoculated with *Achromobacter sp.*, *B. cereus*, *B. halotolerans*, *B. mojavensis*, *P. frederiksbergensis*, and the strains *V. paradoxus* and *P. peoriae* caused the highest increase in chlorophyll content of over 127% for each strain. Chlorophylls are the principal photosynthetic pigment harvesting and converting light energy, and its enhanced synthesis improves photosynthetic rate and PSII efficiency [84]. Enhancement of photosynthesis in plants by PGPR was reported by Sati et al. [85].

A significant reduction in leaf chlorophyll content of more than 50% was observed in plants inoculated with *P. aeruginosa*, which also induced a significant increase in other growth-related variables (shoot and root biomass). This effect can be attributed to a complex interaction of several factors, or trade-offs, in this case between the physiological processes of defense and growth. Activation of defense genes can lead to the production of secondary metabolites, including phytoalexins and pathogenesis-related proteins, which may divert resources away from chlorophyll synthesis, so leading to reduced chlorophyll content [86]. Hormonal crosstalk, including jasmonic acid- and salicylic acid-dependent pathways, appears to play a major role in regulating these tradeoffs, and is modulated by bacterial endophytes. If a bacterial strain primarily activates jasmonic acid-mediated defense responses, this can lead to suppressed growth and

| | Phosphate solubilization | Zinc Solubilization efficiency (SE) % | Indoles-related Production ($\mu g.mL^{-1}$) | Ammonia Production ($\mu M.mL^{-1}$) | HCN production |
|---|---|---|---|---|---|
| *Achromobacter sp.* | + | - | 8.23±1.09[de] | 8.44±0.48[ab] | ++ |
| *Bacillus cereus* | + | - | 9.53±0.74[cd] | 7.25±0.11[bcd] | +++ |
| *Bacillus halotolerans* | - | - | 11.93±3.5[cd] | 7.61±0.87[abcd] | - |
| *Bacillus megaterium* | + | 254%±8,6[bc] | 11.90±0.9[bc] | 6.30±0.41[abcd] | +++ |
| *Bacillus Mojavensis* | - | - | 7.60±0.25[ef] | 8.83±2.49[bcd] | + |
| *Bacillus subtilis* | + | 311%±40,1[ab] | 6.01±1.15[efg] | 4.57±0.43[def] | +++ |
| *Enterobacter hormaechei* | + | 295%±18,9[ab] | 10.07±0.6[cd] | 7.78±0.12[abc] | +++ |
| *Paenibacillus peoriae* | - | - | 2.87±0.57[fgh] | 1.78±0.07[ef] | ++ |
| *Paenibacillus polymyxa* | - | - | 2.03±0.49[h] | 0.86±0.26[f] | + |
| *Pantoea agglomerans* | + | 173%±11,6[c] | 37.40±3.90[a] | 6.86±0.76[bcde] | + |
| *Pseudomonas aeruginosa* | + | 175%±21,9[c] | 22.80±6.3[ab] | 7.81±0.56[abc] | - |
| *Pseudomonas Frederiksbergensis* | + | 368%±23,7[a] | 23.47±0.5[ab] | 9.24±2.51[abc] | +++ |
| *Rhizobium sp.* | - | - | 2.84±0.62[gh] | 9.61±2.00[abc] | ++ |
| *Variovorax paradoxus* | - | - | 15.37±5[bc] | 12.87±1.67[a] | ++ |

**Fig 3. Plant growth-promoting (PGP) traits of the root endophytic bacterial strains. (A)** The symbols "+" and "-" correspond to a semi-quantitative assessment of the recorded activities ("+", positive; "-" negative activity). Data correspond to the mean of a minimum of three biological replicates ± SE. Different letters above data show significant differences within rows ($p \leq 0.05$). HCN, hydrogen cyanide. **(B)** Venn diagram representing the number of species showing positive results for different PGP traits (5*, number of species showed positive results for all the PGP traits). The variables for quantifying the measured biochemical activities are detailed in S2 Fig.

chlorophyll synthesis [87]. Another factor could explain this effect, namely the differential responses of the host toward the bacterial endophyte. Each plant-bacteria interaction is unique, and the response of one particular plant to a specific bacterial strain may differ from another's. The expression of defense- and growth-related genes can vary according to the plant's genetic makeup, its developmental stage, and environmental conditions [88]. To gain deeper insights into these mechanisms, in-depth molecular studies may be necessary to analyze the signaling pathways involved in the plant's defense response and the specific interactions between endophytes and the plant. This approach would help provide a better understanding of the effects of bacterial endophytes on defense responses and plant growth and identify the underlying mechanisms responsible for the reduction in chlorophyll content.

The nitrogen balance index NBI is the ratio of chlorophylls to epidermal flavonols (Chl/Flav) [89], an informative indicator for crop growth. Measuring NBI allows rapid monitoring of both high and low nitrogen status, providing farmers with accurate information to make timely N management decisions. In our study, the NBI values correlated with the growth stimulatory effect of the tested bacterial strains. A significant increase in NBI was noted for plants inoculated with *P. peoriae*, *P. frederiksbergensis* and *V. paradoxus* (Fig 2C) indicating a high nitrogen content and a high nitrogen use efficiency [90].

Another strategy used by the endophytic bacteria to stimulate plant growth is ammonia production. This beneficial metabolite can help meet the plant's need for nitrogen [91–93]. Interestingly, our results showed that all the 14 bacterial endophytes were able to produce ammonia. This production ranged from low production at about 0.86 $\mu M\ mL^{-1}$ for *P. polymyxa* to high production at about 12.87 $\mu M\ mL^{-1}$ for *V. paradoxus* (Fig 3A). This finding is in line with Alkahtani et al. [94], who reported the ability of endophytic bacteria to produce ammonia using the Nessler's reagent method.

The beneficial effects observed in our study align with other direct mechanisms such as phosphate solubilization. Phosphorus is an essential nutrient for plant growth. This macronutrient is present in soil as phosphate, usually as complexes with calcium, iron and aluminum, making it relatively unavailable to plants [94]. Endophytic bacteria have the ability to solubilize inorganic phosphate, making it available for plant uses through the production of

organic acid anions [95, 96]. Our analysis showed that eight bacterial strains out of the 14 isolates could solubilize tri-calcium phosphate forming clear zones on Pikoviskaya agar media (Fig 3A and S2 Fig). Similar studies have reported that the same bacterial species as found positive for phosphate solubilization in our study are able to release accessible forms of phosphate [97–102].

In the same context of enhancing plant nutrient uptake, *in vitro* screening on minimal media containing an insoluble form of Zn, namely its oxide (ZnO), was conducted to test the presence of zinc-solubilizing bacteria (ZSB) among the 14 bacterial endophytes. Zn is a vital micronutrient for cell life. About 10% of enzymes require zinc as a cofactor and its deficiency leads to many metabolic disorders [103, 104]. Several approaches are used in agronomical practices to improve uptake of Zn. ZSB have been selected as natural biofortification agents to amplify Zn concentration, offering a new dimension in crop improvement program to mitigate Zn deficiencies [105, 106]. Zinc solubilization potential of the bacterial isolates was assessed by determining the solubilizing efficiency (SE) in plate assays. Among the 14 isolates, six strains were able to solubilize ZnO. The efficiency of these bacterial strains was checked based on colony and halozone surface. Among them, *P. frederiksbergensis* showed the highest SE of about 368%. This was significantly different from *B. megaterium*, *P. aeruginosa* and *P. agglomerans*, which showed lower SE value of 254%, 175%, and 173% respectively (Fig 3A). Our results are consistent with those of many studies showing the role of these endophytic bacterial species in zinc solubilization [107–110].

One last trait assessed in the present study that could be indirectly related to the observed plant growth-promoting effect, is the production of volatile toxic hydrogen cyanide (HCN). Almost all the isolates (12 isolates out of 14) were positive for HCN production *in vitro*. *B. megaterium*, *E. hormaechei*, *B. subtilis*, *B. cereus* and *P. frederiksbergensis* exhibited the highest HCN production as indicated by a very deep red color on the filter paper (Fig 3A and S2 Fig). HCN is one of a number of chemical substances released by bacteria that can combat plant diseases. Its toxicity is thought to be linked to its reactivity in the electron transport chain, leading to cell death [111]. In addition, HCN secretions seem correlated with the priming of the root length and root hair germination, interacting with the phytohormonal network of organogenesis and plant growth [112]. The result obtained, consistent with other findings [113], suggests the possible use of our HCN-producing endophytes as biofertilizers and biocontrol agents in plant disease suppressiveness [112, 114].

To select the best isolates among the endophytic strains evaluated with high plant growth-promoting potential, a scale was generated and used to assess the PGPR traits. In this scale, the maximum score was 13, considering 1 point for each positive result among the five *in vitro* tests and 1 point for each significant increase among the eight growth variables measured *in planta*. *In vitro*, the following five strains showed high scores for all the PGP traits (Fig 3B): *P. frederiksbergensis*, *P. agglomerans*, *E. hormaechei*, *B. subtilis*, and *B. megaterium*. *In planta*, three different strains (*B. mojavensis*, *P. peoriae* and *V. paradoxus*) obtained six points out of 8, confirming that outcomes depend on several factors, such as the environment, and vary among bacterial isolates. Endophytic impacts on plant performance cannot therefore be easily inferred from simplified *in vitro* assays, although they do offer relevant leads -and very valuable mechanistic knowledge- which will need to be validated by field trials. All in all, *P. frederiksbergensis* was the most effective strain showing the highest value of 10 points. This species has been reported to enhance wheat seed germination and to induce different degrees of wheat seedling growth including shoot and root weight and height, confirming its potential as a biofertilizer for wheat crops [107].

**Biological control activity of the bacterial endophytes against *Fusarium culmorum*.**
*Direct antagonism*. Recent research on endophytes has demonstrated their ability to strengthen the host's defense against diseases and mitigate damage caused by pathogenic microorganisms [115, 116]. The ability of endophytes to colonize the internal tissues of plants would be a significant advantage over other biocontrol agents in providing protection for plants against many pathogens. Endophytes are accordingly the subject of ongoing studies [117]. To assess this antagonistic activity, the most commonly used strategies are *in vitro* direct plate antagonistic reaction against pathogens or comparing the survival rate of inoculated plants with controls. However, knowledge of endophytes in general, their interactions with plant pathogens, and endophyte-induced plant regulation, is far from complete. In this study, we investigated the antagonistic activity of the 14 root endophytic bacterial isolates against *F. culmorum*, the causal agent of fusarium foot and root rot (FFRR), and seedling and head blight diseases (FSB and FHB). The evaluation included direct confrontation tests on PDA medium to assess the inhibition of *F. culmorum* growth by the bacterial isolates.

In dual assays allowing exchange of both volatile and diffusible metabolites, out of the 14 root-associated bacterial isolates tested, nine strains demonstrated antagonistic activity against *F. culmorum*. The inhibition rates ranged from 4% to 66%, indicating significant variations in the effectiveness of the isolates in suppressing *F. culmorum* growth. Among these strains, *P. peoriae*, *P. polymyxa*, *P. agglomerans*, *P.* aeruginosa, and *B. cereus* had effective antagonistic activity of over 40%, stable over time (Fig 4A and S3 Fig). They were selected for the other *in vitro* assays. The antagonistic activity displayed by the selected bacterial endophytes against *F. culmorum* highlights their potential as biocontrol agents. This behavior suggests an antibiosis mechanism, implying the production of active fungicidal compounds, which may be volatile and/or diffusible in the environment. Interestingly, endophytic bacteria have been reported to produce various volatile organic compounds (VOCs) that have a significant role in the control of a broad range of phytopathogens and may be involved in the observed antagonistic effect observed by us [118–120].

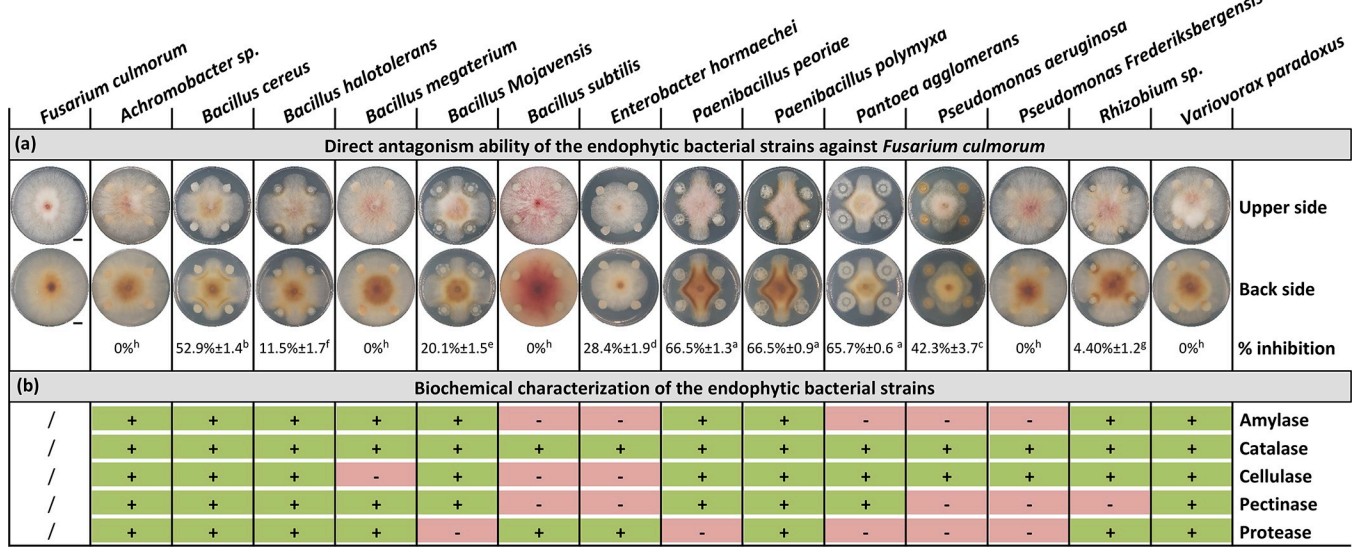

**Fig 4.** Effect of root endophytic bacterial strains on *Fusarium culmorum* mycelial growth in dual confrontation assay (A), and qualitative assay of extracellular enzyme production (B). Dual confrontations and biochemical analyses were recorded after 5 days of incubation at 28˚C. Data correspond to the mean of a minimum of four repetitions ± SE. Different letters above data show significant differences within rows ($p \leq 0.05$). The symbols "+" and "-" correspond to a qualitative assessment of the recorded activities ("+", positive; "-" negative activity). Scale bars represent 1 cm. An enlarged version of the photographic images is available in S3 Fig.

The ability of endophytic microbes to produce hydrolytic enzymes is commonly associated with their plant growth-promoting activity and antagonistic potential against phytopathogens [121]. In our study, all the 14 root-associated bacterial isolates tested positive for extracellular enzyme production, revealed by visible bubbling and degradation zones surrounding bacterial colonies, demonstrating the presence of at least two out of the five evaluated enzymes (Fig 4B and S3 Fig). These enzymes could be attributed to the successful symbiotic relationship between endophytes and their host plant being used to penetrate plant tissues, improve induced systematic resistance, and degrade pathogen cell walls [122]. All the strains were catalase-positive. This enzyme is the first defense line used by microbes to scavenge toxic free radicals generated from biotic and abiotic stress, so indirectly promoting plant growth [58]. The lytic enzymes tested here, such as protease, cellulase, and amylase, may be involved in the degradation of the fungal cell wall and the lysis of the fungal mycelia, while pectinase has been reported to reduce pathogenesis in plants [123].

The production of extracellular enzymes is a process shared between antagonistic and non-antagonistic strains. This suggests that enzymatic activity may not be the sole determinant of antagonistic potential against *F. culmorum*, and that it requires a combination of other mechanisms (e.g., antibiosis and competition for nutrients or space) [117, 124]. In contrast, the simultaneous presence of enzymatic activity and antagonistic potential in certain bacterial strains suggests a multifaceted approach to biocontrol. These strains may use a combination of enzymatic degradation and the production of antifungal compounds to inhibit the growth and development of *F. culmorum*. The synergistic effects of enzyme activity, antifungal metabolites, and volatile organic compounds may enhance their biocontrol efficacy. Further work is need to elucidate the specific mechanisms by which enzymatic activity contributes to antagonistic potential. This could involve exploring the correlation between enzyme activity levels and the inhibition of *F. culmorum* growth, and identifying the specific enzymes and metabolites responsible for the observed effects.

The selection of *P. polymyxa*, *P. peoriae*, *P. aeruginosa*, *P. agglomerans*, and *B. cereus* as the most effective isolates provides promising candidates for further investigation. These isolates demonstrated inhibition rates exceeding 50%, indicating their strong antagonistic potential against *F. culmorum*. Several studies have provided insights into the antagonistic activity of our selected bacterial strain against *F. culmorum*: *P. polymyxa* [125], *P. aeruginosa* [126], *P. agglomerans* [127], *B. cereus* [128], but to the best of our knowledge, ours is the first thorough evaluation of the antagonistic potential of *P. peoriae* against *F. culmorum*.

*Inhibition of Fusarium culmorum sporulation*. Sporulation plays a crucial role in the life cycle of *F. culmorum*, and the highly effective spore dispersal ability of this fungus accounts for its exetensive dissemination [129, 130]. *F. culmorum* produces asexual spores such as macroconidia and chlamydospores that form the most efficient means of reproduction and the main source of plant contamination [131]. Investigating the impact of the selected bacterial strains on *F. culmorum* sporulation can therefore provide valuable insights into their potential as biocontrol agents for controlling spore propagation and infection.

Our study of the inhibition of *F. culmorum* macroconidia formation assay showed that all strains had an effect on the formation of macroconidia. In the presence of *B. cereus*, *P. agglomerans*, *P. peoriae*, and *P. aeruginosa*, *F. culmorum* macroconidia were completely absent, indicating total inhibition of *F. culmorum* sporulation. To a lesser degree, *P. polymyxa* caused partial inhibition, significantly reducing by 56.62% the number of the macroconidia formed (Fig 5).

The absence of macroconidia in the presence of *B. cereus*, *P. agglomerans*, *P. peoriae*, and *P. aeruginosa* and the reduction in sporulation induced by *P. polymyxa* suggest that these bacterial strains exert strong inhibitory effects on the spore formation of *F. culmorum*. This

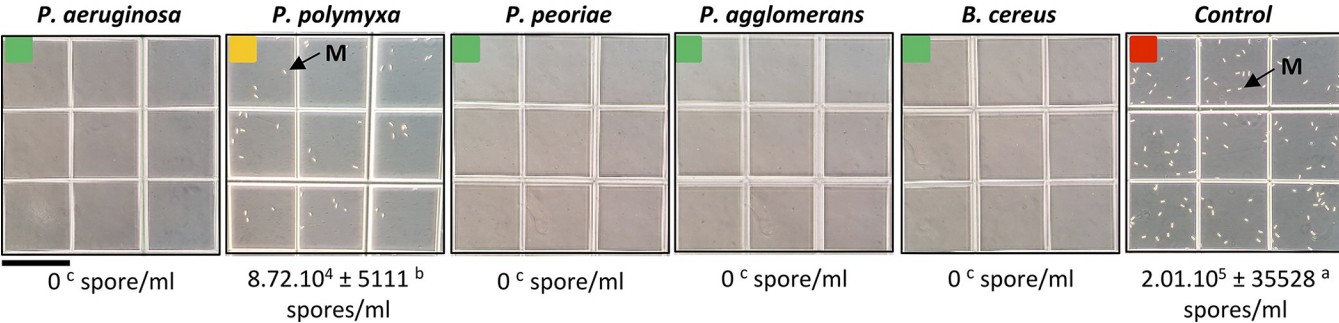

**Fig 5. Optical microscopic observation of *Fusarium culmorum* macroconidia formation after 30 days (from dual culture assay of the five selected bacterial strains).** Green square indicates total inhibition of *F. culmorum* macroconidia production, yellow square indicates partial inhibition and red square indicates macroconidia formation in the control; scale bar = 0.33 mm; M: *F. culmorum* macroconidia. Data correspond to the mean of a minimum of three biological replicates ± SD. Different letters above data show significant differences within rows ($p \leq 0.05$).

inhibition could be due to the modification of the environmental conditions that trigger sporulation, including pH, nutrient depletion, oxidative stress, and toxic secondary metabolites (here with antigerminative activities) produced by the competing endophytes [131]. Interestingly, the partial inhibition of *Paenibacillus polymyxa* could be due to differences in intensity of the antagonistic interaction between *P. polymyxa* and *F. culmorum*, underlining the plurality of the interactions between rhizobacteria and plants.

**Bacterial culture filtrates inhibited F. culmorum mycelial development.** The antifungal activity of the culture filtrates of the five selected bacterial strains was further evaluated. The results showed that all bacterial culture filtrates differentially inhibited *F. culmorum* mycelial growth. The rate of inhibition ranged from 10.38% for *P. aeruginosa* to 47.04% for *P. polymyxa*, which exhibited the highest inhibition rate in the dual culture assay. *F. culmorum* generally exhibits a moderate growth rate and produces a dense cottony mycelium that covers the surface of the agar medium with a smooth outer surface. In the presence of the bacterial culture filtrates, *F. culmorum* showed distinct changes in growth characteristics compared to controls. The colonies appeared thinner, with clear irregular edges, indicative of compromised mycelial density; the aerial mycelium was sparsely distributed, and the overall growth rate was notably reduced (Fig 6).

The observed inhibition of *F. culmorum* mycelial growth by bacterial culture filtrates suggests the presence of active substances with antifungal properties. The differential rates of

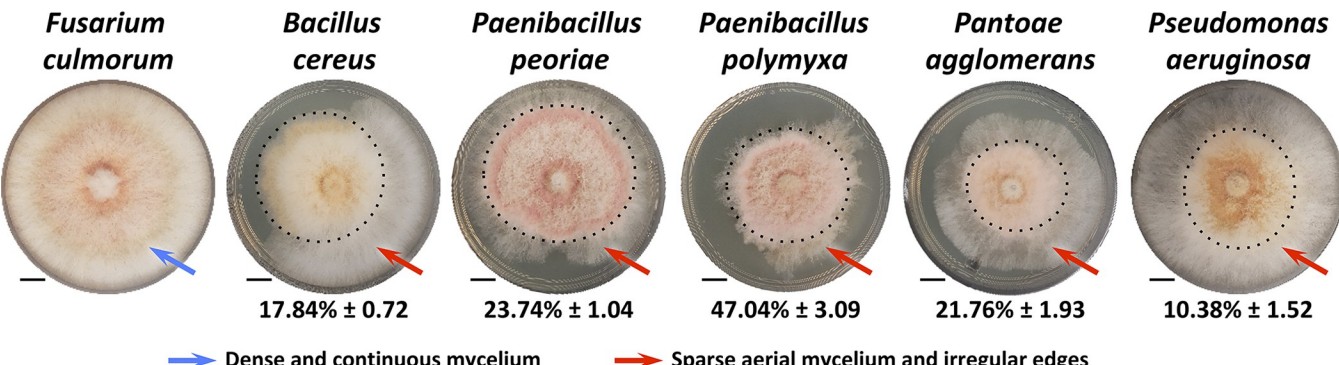

**Fig 6. Effect of culture filtrate of selected bacterial strains on *F. culmorum* mycelial growth after 7 days of incubation at 28°C.** Inhibition of the mycelial growth is expressed in % ± SE. Morphological differences between mycelium are highlighted by different colored arrows. Scale bars on photographs represent 1 cm.

inhibition indicate variations in the composition and/or concentration of these active substances among the bacterial strains. These data are in line with some reports indicating that the capacity to produce and secrete bioactive metabolites may differ even between bacterial strains belonging to the same species [132].

The secretion of hydrolytic enzymes and antimicrobial metabolites are among important traits associated with most endophytic microorganisms essential to the control of phytopathogens. These enzymes are very diverse, and some families, such as cellulases, proteases, or pectinases, have extracellular members reported to be correlated with the antifungal activity of *Bacillus simplex* and *B. subtilis* against some *Fusarium* species, resulting in fungal hyphal thinning [121, 133].

Our results revealed similar activities with the five selected species that showed *F. culmorum* mycelial inhibition producing at least one of the above enzymes (Fig 4B). The changes in *F. culmorum* growth characteristics further emphasize the impact of bacterial culture filtrates on fungal development. These marked adverse physiological effects indicate disruptions in mycelial growth, cellular integrity, and overall fungal fitness in the presence of the active substances present in the bacterial culture filtrates. *Bacillus* strains are known to produce a broad spectrum of antimicrobial compounds. Lipopeptide biosurfactants (including surfactins, iturins, and fengycins) are the main class of antibiotics, playing essential roles in inhibiting plant pathogens growth [134]. Wang et al. [135] found fengycins in particular to show significant antifungal activities against *Fusarium graminearum*, causing morphological changes in fungal cell walls and membranes [136, 137]. The effectiveness of *P. aeruginosa* relies on the production of diverse antifungal compounds such as chitinases, glucanases, 2,4-diacetylphloroglucinol (DAPG), pyoluteorin, phenazines, siderophores, and surfactants [138–140]. Similarly, *Paenibacillus* species have drawn much attention by its capacity to synthesize biological control substances including lipopeptide antibiotics, bacteriocins, and antibacterial proteins [141]. Previous research also found that *P. agglomerans* exhibited biocontrol potency, being able to produce bioactive lipopeptides inhibiting fungal growth [142]. These substances will occur in the cell-free filtrates, suggesting the sensitivity of *F. culmorum* to the diffusible metabolites produced by the bacterial endophytes during dual culture assay. The identification of these substances in future studies will be of great interest.

**Inhibition of F. culmorum macroconidia germination by bacterial cells and intracellular metabolites.** To characterize the antifungal potential of these strains besides mycelial growth and sporulation, their potential to interfere with macroconidia germination was assessed by co-inoculating *F. culmorum* macroconidia with bacterial cells allowing direct interaction and physical contact.

Spores are essential in the *Fusarium* infection process and disease spread. Spore germination is known to be the first step in fungal mycelium development. Understanding the inhibitory activity of the selected bacterial endophytes on spore germination in therefore pertinent.

Visually, all bacterial strains, expect for *B. cereus*, inhibited the germination of *F. culmorum* macroconidia on PDA medium, as evidenced by the absence of any visible fungal colonies (Fig 7A). However, microscopic examination revealed that three of these visually inhibitory strains completely suppressed germination, as indicated by the absence of any macroconidia germination. The strain *P. agglomerans*, despite visual inhibition, allowed germination at the microscopic level, but it was limited, with sparsely distributed hyphae showing abnormal characteristics, including hyphae swelling and chlamydospore formation (Fig 7B).

The complete inhibitory effect observed for *P. peoriae*, *P. polymyxa*, and *P. aeruginosa* is indicative of their potent antifungal activity against *F. culmorum*. These strains effectively prevented the germination of macroconidia and the emergence of germination tubes. This total inhibition suggests that these bacterial strains possess powerful mechanisms that suppress key

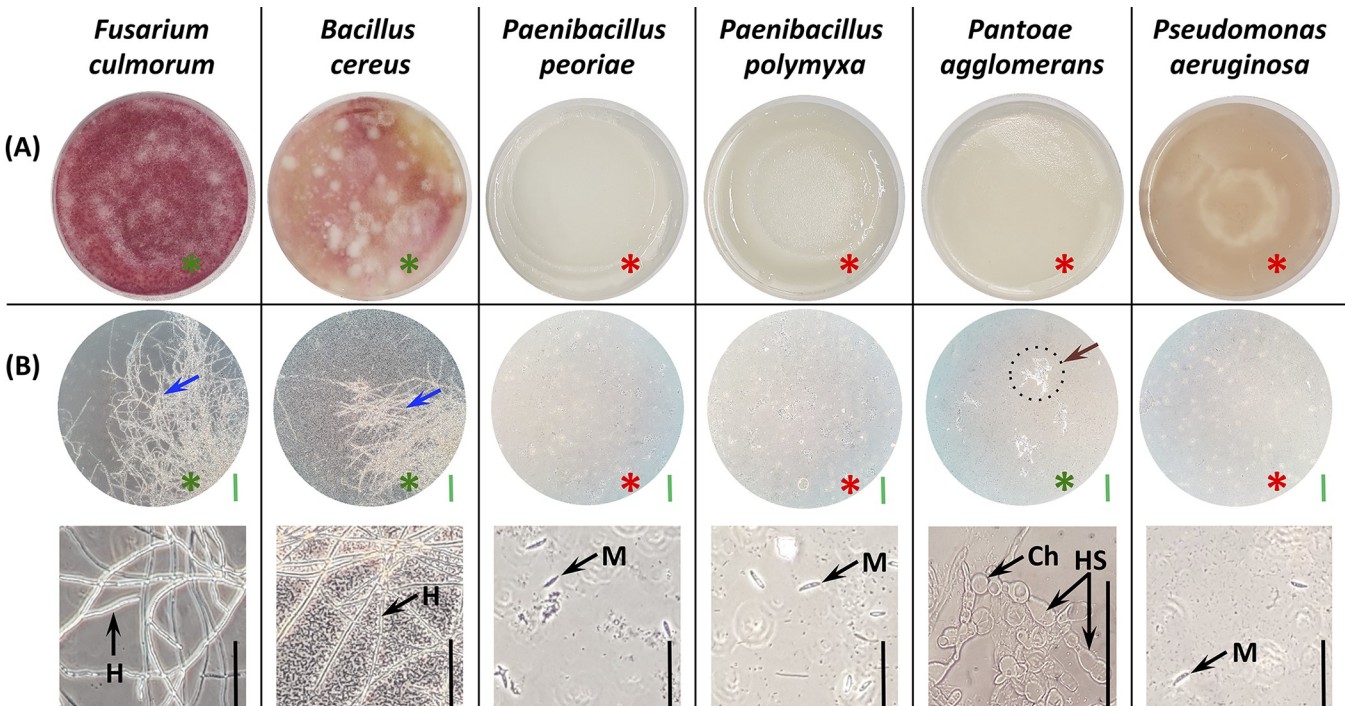

**Fig 7. *F. culmorum* macroconidia germination in the presence of the selected bacterial strains after 5 days of incubation at 28°C. (A)** Macroconidia germination appearance on PDA (Petri dish surface side, Ø 8 cm). (**B**) Optical microscopic observation of macroconidia and hyphae. Green scale bar = 0.33 mm; black scale bar = 0.1 mm. Red asterisk indicates no germination; green asterisk indicates visible germination at visual and microscopic levels. Blue arrows for a dense mycelium; brown arrow for a delayed and less well-developed mycelium. **M**: macroconidia, **Ch**: chlamydospore, **HS**: hyphal swellings, **H**: hyphae.

steps in the germination process of *F. culmorum*. The prevention of macroconidial germination indicates that they interfere with the activation of metabolic processes or signaling pathways necessary for germination initiation. This inhibition of germination may be mediated by the production of substances that disrupt cell division or affect the integrity and elongation of the emerging hyphae.

*Pantoea agglomerans* exhibits a unique mode of inhibition. Although this strain visually inhibited germination, the presence of sparsely distributed hyphae at the microscopic level indicated partial germination. However, the abnormal characteristics displayed by these hyphae, such as swelling and chlamydospore formation, suggest that *P. agglomerans* exerts inhibitory effects on *F. culmorum* through inducting aberrant fungal growth. The specific mechanisms underlying this inhibition remain unclear and warrant further molecular investigation. The presence of hyphal swelling and the formation of chlamydospores -considered as survival structures- suggests that *P. agglomerans* may trigger a stress response in *F. culmorum*, probably due to specific secondary metabolites or signaling substances that interfere with fungal development. Alternatively, *P. agglomerans* might modulate the expression of fungal genes involved in cell wall biosynthesis or morphogenesis, leading to the observed morphological abnormalities.

Understanding the effects of bacterial intracellular fractions on *F. culmorum* spore germination and hyphal development is crucial for elucidating the mechanisms underlying their antagonistic activity.

Spore germination rates varied according to the bacterial strains and time intervals. At 6 hours, the control exhibited a germination rate of 82.7%, whereas this rate was significantly reduced in samples exposed to intracellular fractions of *P. agglomerans*, *P. aeruginosa*, and *B.*

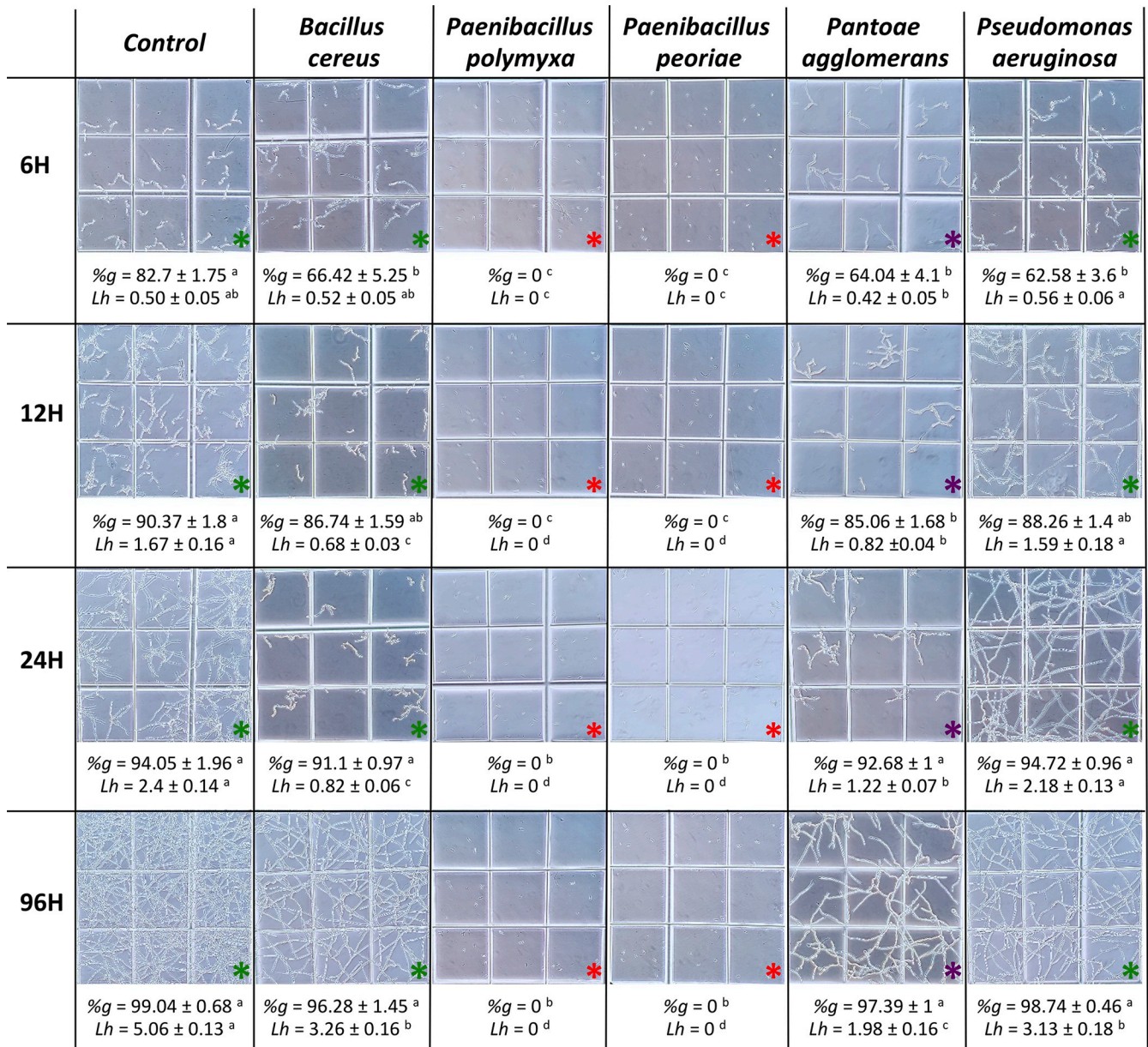

**Fig 8. Microscopic observation of *Fusarium culmorum* macroconidia germination after 6, 12, 24 and 96 h of exposure to intracellular metabolites of different selected bacterial strains at 28°C.** Red asterisks indicate no germination; green asterisks indicate germination and normal hyphal morphology; and purple asterisks indicate germination with misshapen hyphae. "*%g*" corresponds to the percentage of macroconidia germination; "*Lh*" corresponds to length of hyphae (mm) per square (0.033 x 0.033 mm). Scale bar = 0.33 mm. Data correspond to the mean of a minimum of nine repetitions ± SE. Different letters above data show significant differences within rows ($p \leq 0.05$).

*cereus*, with 64%, 63% and 66.4% respectively. *P. peoriae* and *P. polymyxa* intracellular fractions caused a complete absence of spore germination (Fig 8).

At 12 h, similar trends were observed. The controls consistently showed the highest germination rates, and intracellular fractions of *P. agglomerans* exhibited a significant inhibitory effect illustrated by a delayed germination, while *P. peoriae* and *P. polymyxa* intracellular fractions continued to show no germination and no germ tube emergence was detected.

After 24 h and 96 h, no significant inhibition was noted, indicating that the inhibitory effect of the bacterial endophytes was only marked in the first 6 h of germination, except for *P. peoriae* and *P. polymyxa*, which showed complete inhibition and a strong inhibitory effect on germination even 4 days later.

Hyphal development was also influenced by bacterial intracellular fractions when germination occurred. At 6 h, the intracellular fractions of *P. agglomerans*, *P. aeruginosa*, and *B. cereus* showed no activity on mycelial growth. However, at 12 h and 24 h, the control consistently displayed the longest average hyphal lengths, while the intracellular fractions of *P. agglomerans* and *B. cereus* exhibited significant shorter lengths. After 96 h, the *P. agglomerans* intracellular fraction showed the highest inhibitory effect on mycelial growth, with less than 2 mm of mean hyphal length (per square), and significant reductions were obtained with *B. cereus* and *P. aeruginosa* (Fig 8).

Microscopic observations revealed distinct effects of the bacterial intracellular fractions on *F. culmorum* macroconidia and hyphal morphology (Fig 9). The intracellular fractions of *P.*

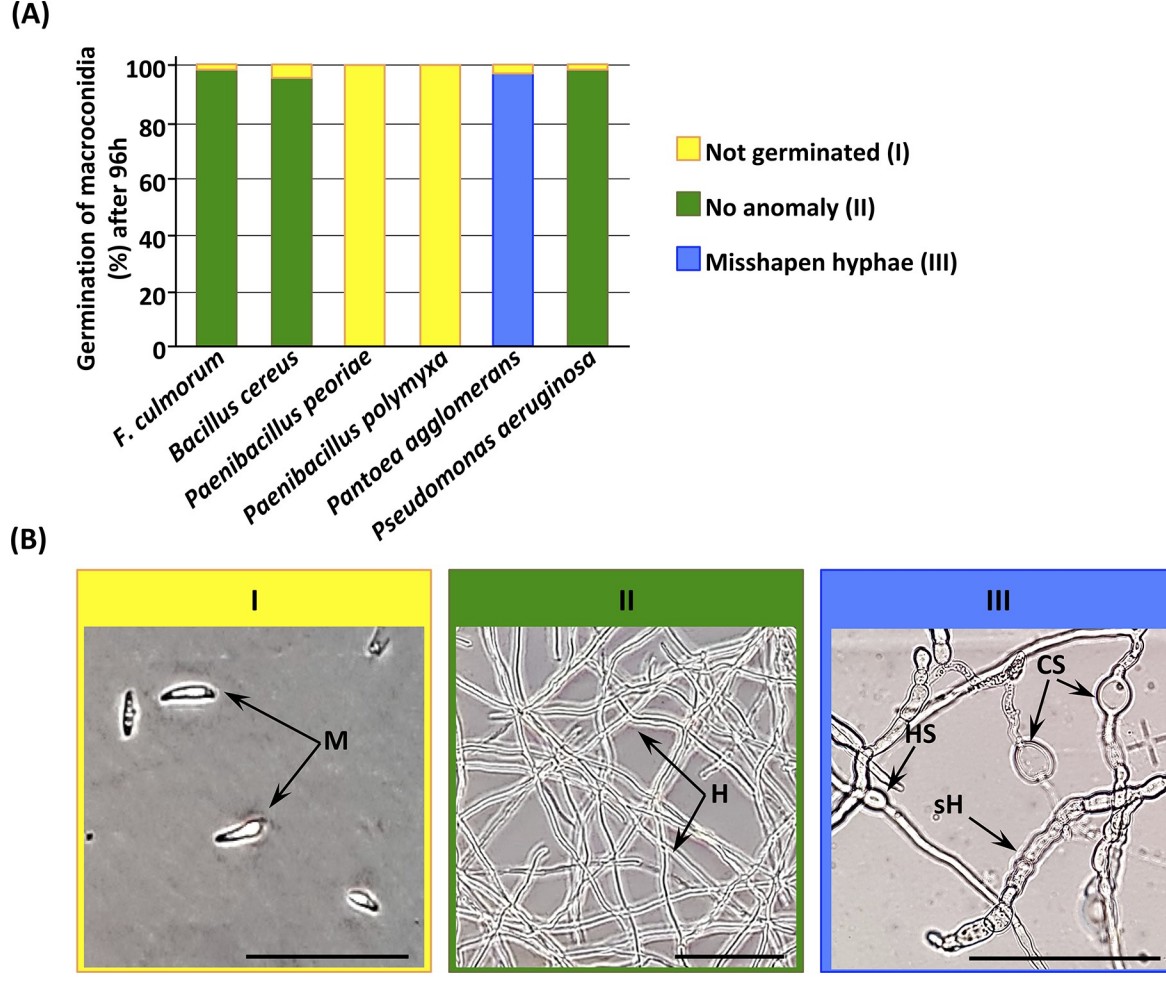

**Fig 9. *F. culmorum* macroconidia germination and related cell structures after 96 h of exposure to rhizobacteria's intracellular metabolites.** (**A**) Rate of *F. culmorum*'s macroconidia germinated after exposure of intracellular metabolites of the five selected rhizobacteria. (**B**) Example of the microscopic morphologies of macroconidia, mycelia, and chlamydospores of *F. culmorum*: **I**, not germinated; **II**, no anomaly; **III**, misshapen hyphae. **M**, macroconidia; **H**, normal hyphae; **sH**, septate hyphae; **HS**, hyphal swelling; **CS**, chlamydospores. Scale bars = 100 μm.

*polymyxa* and *P. peoriae* resulted in non-germinated macroconidia, showing their strong inhibitory effect on germination (Fig 9A and 9B).

Members of the *Paenibacillus* genus are able to produce antimicrobial lipopeptides. This family of secondary metabolites with broad-spectrum antimicrobial activity includes polypeptins, iturins, polymyxins, surfactins, and fusaricidins [143–146]. Fusaricidin produced by *P. polymyxa* has been reported to inhibit spore germination and disrupt hyphal membranes of *Fusarium oxysporum* f. sp. *cucumerium* [147].

Conversely, the intracellular fractions of *P. aeruginosa* and *B. cereus* had no impact on hyphae, resulting in normal hyphal thickness comparable to controls. In contrast, the intracellular fraction of *P. agglomerans* led to misshapen hyphae characterized by swelling, septation, and the presence of chlamydospores (Fig 9A and 9B). These abnormal hyphal characteristics indicate an impairment in *F. culmorum*'s normal growth pattern and suggest that this strain exhibits fungicidal rather than fungistatic activity. Our finding is consistent with similar results demonstrating that *Pantoea dispersa* strongly inhibited the pathogenic fungus *Ceratocytis fimbriata* growth and spore germination causing dramatic breakage of its hyphae [148]. *Pantoea* species are known to be strong environmental competitors producing a wide range of naturally occurring substances with antibiotic activity such as phenazines, pantocins, and microcins [149, 150].

Bacterial intracellular metabolites of *P. polymyxa*, *P. peoriae*, and *P. agglomerans* produced similar effects when applied as cell suspensions, suggesting the presence of the same or similar active compounds. The observed inhibition could also be mediated through secreted products, whose identity was not determined in this study. In the case of *P. aeruginosa*, the full inhibition of macroconidia germination was achieved only by the bacterial suspension, indicating that this total inhibitory effect requires direct cell-cell interactions.

As fungal germination depends on environmental pH and availibility of nutrient such as carbon and nitrogen, total inhibition or reduced germination and hyphae development could be related to the modulation of these variables and/or nutrient uptake directly by bacterial cells that may enter into a competitive relationship and/or through their intracellular metabolites [151]. This result suggests that the intracellular metabolites of *P. polymyxa*, *P. peoriae*, and *P. agglomerans* characterized by their high effectiveness could be developed as potential biocontrol agents for *F. culmorum* diseases. However, the specific substances responsible for the observed inhibitory effect remain unknown, pending further research.

## Root endophytes as molecular inducers of wheat growth and defense responses

**General considerations.**   Plant-microbe interactions dynamically affect both plant growth and health. The mechanisms underpinning these associations are primarily mediated by specialized host-derived metabolites (or elicitors) that can trigger rapid and extensive transcriptomic reprogramming. The plant response promotion experiments we conducted here through real-time PCR analyses revealed that seed inoculations with each endophyte rhizobacteria significantly modulated, 28 days after treatment, all selected genes involved in plant growth and defense accumulation in roots and leaves (Fig 10 and S2 Table). These transcriptomic responses were characterized by inductions or inhibitions, which occurred over time post-inoculation, and systemically throughout the plant (under- and aboveground organs). In addition, the gene expressions seemed to be strain-dependent, with differential patterns that were intra-clade-dependent (e.g., *Bacillus*, *Pseudomonas* and *Paenibacillus*). These transcriptomic responses highlight the complexity and richness of the interactions established between wheat root endophytes and wheat, which ultimately

**Fig 10. Quantitative real-time RT-PCR of selected genes related to plant growth (growth modulators) and innate immunity (recognition and transduction of signals, pathogenesis-related proteins, secondary metabolisms, oxidant stress and fluxes, phytohormone balances).** Transcript abundances were recorded in leaf and root tissues of wheat (cv. Karim) at 28 days after seed inoculation with PGPR ($10^8$ CFU / mL). Transcript abundances of the genes were recorded by real-time $q$RT-PCR analyses in the PGPR-infected plants and were relative to those of the control using the $2^{-\Delta\Delta CT}$ method. The significant differences were statistically analyzed based on three independent biological replications. Each biological repetition corresponds to a mix of five plants (Tukey's test, adjusted $p < 0.05$). Complete statistical data, including the SE, are detailed in S2 Table. Colors represent level of transcriptional expressions (blue: up-expression; red: down-expression).

determine a myriad of molecular mechanisms involved in various physiological properties such as plant growth and innate immunity.

**Wheat growth-dependent molecular responses.** PGP microbes directly enhanced plant growth by facilitating solubilization, mineralization and nutrient uptake, as well as the expression of a range of plant growth regulators and signaling molecules related to phytohormone production (e.g., abscisic acid-ABA, gibberellic acid-GA, indole-3-acetic acid-IAA, and, for present purposes, cytokinins-CK) [152]. CKs stimulate cell division and enlargement and shoot growth, and they induce stomatal opening under stress conditions (salinity or drought); they also play a vital role in delaying premature leaf senescence and death, and in enhancing plant stress tolerance [153, 154]. Two genes related to the CKs-dependent signaling pathway seem to be involved in plant growth: the first, *IPT5*, encodes an adenosine phosphate-isopentenyltransferase isoform that catalyzes the synthesis of cytokinins by attaching an isopentenyl radical to the purine ring of adenine, and the second, *CKX*, encodes an isoform of cytokinin oxidase that catalyzes the irreversible oxidative degradation of CKs [155]. Our transcriptomic data showed that *IPT5* was induced (except with *P. polymyxa*, where it was slightly inhibited in both organs), whereas *CKX* was not significantly modulated or repressed, especially with *B. halotolerans*, *Rhizobium*, and *Variovorax*. These expression kinetics could be explained by the enzyme function encoded by these genes, and in this regard, *IPT5* and *CKX* profiles are consistent with the potential of plant growth promotion developed by the endophytes (Fig 1).

**Wheat innate immune-dependent molecular responses.** Plant defense responses are driven by a complex coordinated induction of heterogeneous groups of phytohormones such as SA-, JA-, ET-signaling pathways, and transcription factors, which collectively orchestrate the cellular integration of environmental cues by initiating the local and systemic release of indigenous defense-related enzymes and secondary metabolites.

The crosstalk between the different hormonal networks is complex and literature reports are sometimes contradictory: the JA and ET signaling pathways are generally described as synergistic with each other, and antagonistic to the SA-dependent pathway. In addition, the SA pathway may be responsible for the systemic acquired resistance (SAR) and deliver resistance against biotrophic pathogens, and the JA/ET pathways may be responsible for the induced systemic resistance (ISR) and be associated with symbiont or necrotrophic pathogens [156]. Besides constituting the central regulatory network of plant immunity, these phytohormones interact with growth-related plant hormones, including cytokinins (CKs), themselves involved in the promotion of plant resistance to pathogens [157]. The modulation (including up- and/ or down-regulation) of interconnected phytohormonal pathways is therefore likely to be a significant part of an overall effect of individual root endophytes on innate immune -and growth-performance we outline here.

Root endophytes engage several hormonal-signaling pathways to regulate the elicitation of plant disease resistance against a wide range of parasite- and pathogen-induced challenges [158]. In our molecular study, the modulation of defense responses observed in leaves and roots following the inoculation of the 14 root endophytes was coupled with the increased expression of various generic stress-related transcription factor (*WRKY3*, *EDS1*), of SA- and JA-responsive genes *NPR1* and *AOS*, respectively, together with reactive oxygen species (ROS)-associated genes *CAT*, *SOD* and *POD*. In addition, CKs may well be involved in the induction of the defense gene expression (PR-proteins, phytoalexin synthesis, and lignification processes), through the regulation of the SA-signaling pathways [158, 159]. Nevertheless, the isolates suppressed, in leaves and roots, the expression of genes involved in the ethylene biosynthesis (aminocyclopropane (ACC) synthase (*ACS1*)), and the TF pathogen-induced ERF1 (*PIE1*) related to the ET-signaling pathway, which regulates the expression of second-order ET-dependent regulatory genes. Our data indicate that the strains upregulated the CKs, SA, and JA signaling pathways, but repressed the ethylene pathway. Synergistic overlap of SA-, JA-, and CK-signaling pathways seems to occur here, as is well described for some pathosystems. However, these patterns can be confusing, because SA- and JA/Et-dependent signaling pathways are often described as antagonistic. The role of each hormone, and their relationship to each other during relatively long time periods in the plant development cycle under the influence of beneficial microbial agents, is still not well understood. The synergistic action of SA- and JA-signaling pathways has already been described for some pathosystems, as has the suppression of the ET-signaling pathway by the root endophytes [160–162]. The inhibition of ethylene synthesis can be paralleled by another effect: during compatible interaction, it causes cell collapse both during PTI and disease development in plant tissues, thus helping the colonization of the host plant by the pathogens [163, 164].

These complex cellular responses involve timely recognition of the invading pathogen by R proteins. In this regard, and in our study, two receptor-like kinases (RLKs), *WAK2* and *LRR-RLK*, showed abundance increases in leaves, possibly in relation with the systemic signals elicited by the SA-, JA- and ROS-signaling pathways from roots. In roots, *WAK2* was somewhat inhibited, whereas *LRR-RLK* was activated. The biochemical modalities related to their function of recognizing various exogenous factors are different [165, 166], WAK2 and LRR-RLK being described as conferring resistance against *F. culmorum* [167] and *Blumeria graminis* f. sp. *tritici*, respectively [168]. Their contrasting transcriptomic accumulation observed in this study is plausibly related to the relationship that each root endophytes forms specifically with the plant host.

Subsequent plant responses to microbe recognition and phytohormone balance modulation events are associated with profound metabolic adjustments leading to the accumulation of several defensive substances. In the first line of defense induced early in a biotic interaction

(including positive plant-beneficial microorganism interactions), plants produce reactive oxygen species (ROS), *viz.* superoxide anion radicals ($O_2^{\circ-}$), hydrogen peroxide ($H_2O_2$), hydroxyl radicals ($OH^-$), singlet oxygen ($O_2^1$), and alkoxy radicals (ROs). ROS are formed as by-products of various metabolic pathways localized in different cellular compartments. In an interaction biotic context involving pathogens as well as symbionts or beneficial agents, ROS production can be enhanced in the apoplasm, responsible for oxidative stress. Several enzymes are responsible for this massive and sudden ROS production, including the class III peroxidases (POD). PODs are redox enzymes that trigger the biosynthesis and the oxidation of phenolic compounds that integrate the wall-building processes (lignification) of host plant cells during the defense reaction against pathogenic agents. As such, members play an active role in ROS homeostasis in plant-microbe interactions and hold a key position in plant defense against pathogens [169], including in wheat [170]. However, an increase in ROS levels in plants above a certain threshold (also called an oxidative burst), leads to severe disruption of proteins, peroxidation of membrane lipids, which results in decreased fluidity and fracturing of the membranes, and DNA/RNA damage. The combination of all these spontaneous chemical reactions impairs the normal functioning of plant cells, leading eventually to cell collapse or hypersensitive reaction (HR) corresponding to incompatible interaction [171]. Plant ROS are powerful weapons against pathogens, and to protect against the non-selective harmful effects of superoxide and $H_2O_2$ (and therefore preventing the initiation of the Haber-Weiss reaction leading to hydroxyl ($OH^-$) radical generation) plants are armed with antioxidant defense systems both non-enzymatic (e.g., ascorbate, glutathione, flavonols, tocopherols) and enzymatic such as catalases (CAT), superoxide dismutase (SOD), and Foyer-Halliwell-Asada (or ascorbate-glutathione) cycle [172]. These enzymes are usually co-expressed and play a crucial role in the mitigation of ROS toxicity. It has been reported that some root endophytes (and PGPRs), including *B. thuringiensis*, *Enterobacter* spp., *P. aeruginosa*, and *Pseudomonas mendocina*, increase the activity of antioxidant enzymes in wheat, tomato, mung bean, and lettuce, respectively. Our results showed a significant increase in *CAT* and *SOD* gene expressions (except for *Rhizobium* sp. in roots), and for *POD*, except for *Achromobacter* sp., *E. hormaechei*, *P. frederiksbergensis*, and *V. paradoxus*, where *POD* modulations were not significant, and with *Rhizobium* sp., which drastically repressed the *POD* gene expressions.

The modulation in the *POD*, *SOD*, and *CAT* expressions indicates the existence of an oxidative stress that is either still ongoing but at very low levels or has been resolved with the enzymatic systems still on alert in roots and leaves. Anyway, Whichever the case, PGPRs induced a systemic strengthened antioxidant system for ROS removal. It is of note that the increased activities of antioxidant enzymes in the root endophyte-inoculated wheat probably enhanced plant growth observed in Fig 2, by eliminating the free radicals and maintaining efficient photosynthetic rates and N uptakes (Fig 2), and plausibly cellular redox potential and membrane integrity. These findings are in close agreement with others reported with various biocontrol agents [173–175].

Another generic stress marker is the heat shock protein 70 (HSP70), upregulated by pathogen-induced signaling pathways, notably in wheat (named *Ta*HSC70) [176, 177]. HSP70 is an evolutionarily conserved family of proteins typically localized in the cytosol and various organelles. Members may function as molecular chaperones, playing roles in protein transport and assembly processes. They may thus be recruited for folding and refolding of nonnative proteins to prevent irreversible aggregation of stress-denatured proteins, notably under the action of ROS produced during the incompatible interactions, thus preserving cellular homeostasis [171, 178]. In our transcriptomic study, *Hsp70* was upregulated, except for the two *Paenibacillus* species and *Rhizobium* sp. in roots, where abundances were downregulated. Interestingly, these modulations correlate with the kinetics of ROS-scavenging enzymes, knowing that

HSP70 plays a crucial role in maintaining cell integrity in a prooxidative context. Furthermore, *HSP70* expression is co-regulated with the expression of genes related to hormonal balances: for instance, HSP70 expression correlates with that of *IPT5* in leaves (but not in roots), which is involved in the CKs-signaling pathway, the *allene oxide synthase* (*AOS*) gene, whose related enzyme is involved in the JA production and whose expressions were upregulated in roots and leaves, and the *non-expresser of pathogenesis related 1* (*NPR1*) gene, whose related proteins are master regulators of SAR induced by SA and JA, and for which transcript accumulations were upregulated in both organs (except for the two *Paenibacillus* species in roots with downregulations). The interactions between the HSPs and most phytohormones and ROS have been extensively studied in many plant species [179], and the expression trends observed here support our hypothesis concerning the HSP family's involvement in the positive interactions that PGPRs initiate with their hosts.

However, there arises the question of the diffusion of ROS, in particular $H_2O_2$. This small molecule is uncharged and relatively long-lived. These specific physico-chemical features enable it to spontaneously cross membranes. In infinitesimal amounts, it is a suitable candidate to provide a secondary messenger role, integrating the complex phytohormonal balances that regulate multiple physiological pathways in plants [180]. This key-mediating role also involves the intervention of specific transmembrane channels, called peroxyporins, whose role is to finely control its diffusion [181]. Peroxyporins are specific aquaporins of the MIP family, such as certain PIPs (plasma membrane intrinsic proteins). However, despite their primordial role in the diffusion of the second messenger $H_2O_2$, peroxyporins have not been studied in PGPR-treated plants. In our study, the increased abundance of *AQP* transcripts (belonging to the PIP group) correlates with those of *CAT*, *POD* and *HSP70*, reflecting a plausible role for PIPs in the root endophyte-induced $H_2O_2$-signaling pathway. All these positive gene co-regulations point to *HSP70* and *AQP* being involved in basal defense through complex interconnections between CK-, SA- JA-, and $H_2O_2$-dependant signaling pathways (as is commonly observed with classical defense markers such as CAT, SOD and POD), which are triggered by the interaction of the wheat plant with the root endophytes, and which persist systemically over time. Finally, these observations confirm that these proteins are an integral part of plant immunity and participate in PTI responses that can be initiated by beneficial microorganisms such as root endophytes and PGPRs, as is also observed with interactions involving phytopathogens [182, 183].

In tandem with pro-oxidant compounds, some root endophytes are known to significantly induce the accumulation of a wide range of specialized host-derived secondary metabolites that are potentially antioxidant and/or toxic to microorganisms, such as phenylpropenes, terpenes, and pathogenesis-related proteins.

Phenylalanine ammonia lyase (PAL) is the primary entry enzyme of the shikimic acid pathway, which regulates biosynthesis of low-molecular-weight phenolic compounds, such as cinnamic, coumaric, and caffeic acids, flavonoids, tannins, and lignins [184]. This molecular family is remarkably diverse and widely distributed in the plant kingdom. It includes the flavonoids, a richly varied group, of which almost all subgroups have the capacity to act as antioxidants and/or biopesticides (or phytoalexins), regulating both plant development and the interaction with microbes regardless of their trophic modalities (commensal, pathogenic or beneficial) [185]. In addition, PAL degrades *L*-phenylalanine to *trans*-cinnamic acid, integrating the biosynthesis of SA to induce the defense responses [186]. In our study, most root endophytes elicited *PAL* expression in roots and leaves, except for *V. paradoxus*, which repressed it in both organs. Correlated with this gene expression, leaf epidermal flavonol accumulation increased slightly in some wheat strains, including *B. megaterium*, *P. agglomerans*, and *Rhizobium* sp. (Fig 2). These molecular ecophysiology data are congruent with many studies that

have demonstrated that flavonoids are key active chemicals that participate in the mediation of communication networks between rhizobacteria and plants [187, 188]. The accumulation of these phenolics maximized the tolerance of wheat against salinity and drought events when inoculated with *Bacillus pumilus*, *Pseudomonas mendocina*, *Arthobacter* sp., *Halomonas* sp., and *Nitrinicola lacisaponensis* [189]. Root endophytes (and PGPRs) play a significant role in initiating the plant's induced systemic tolerance (IST) to various abiotic stresses, which constitutes the basis of eco-friendly stress management to enhance plant tolerance to major environmental cues that are amplified by general climate change (such as salt, drought, and nutrient deficiency or excess). These promising potentials will be further studied in future research on the best performing PGPRs isolated in this work.

Besides these small secondary metabolites, plants synthesize proteins with potential intrinsic toxic properties. This concerns pathogenesis-related (PR) proteins (PRPs) and small antimicrobial peptides. These proteins are small, mostly acidic, resistant to breakdown, and most commonly found in intercellular spaces. The biochemical activities of these markers have been reported to interfere with membrane integrity and/or to degrade cell walls of the pathogens [190]. They are among the major preliminary proteins accumulated as molecular defensive barriers against biotic agents, substantially limiting the degree of pathogenic microorganism invasion and spread throughout the plant. Our transcriptomic analyses show that the isolated rhizobacteria significantly elicited the expression of several of these candidates (*PR-1*, *PR-5*, *POD*, *thionin*). The upregulation of PRPs in tandem with the accumulation of the low molecular-weight phenolic compounds reflect the rhizobacteria's ability to trigger the plant's innate immune performances efficiently and systemically.

Several defense-encoding genes showed more complex regulations, being either weakly modulated or significantly inhibited by some rhizobacteria, in roots and/or in leaves. Full inhibitions were observed with *polyphenol oxidase* (*PPO*, a copper-containing metalloprotein that catalyzes the oxidation of phenolic compounds to quinones, making them more toxic to microorganisms than the original phenolic compounds), and the antimicrobial *LTP* (lipid transfer protein) and *PR-17*. Contrasting modulations between organs were observed (i.e., upregulation *vs*. downregulation), with the *glucan synthase-like 22* (*GSL22*, involved in the callose synthesis), as well as the *glycine-rich protein* (*GRP*, small antibacterial and antifungal proteins) and the two classes of antimicrobial PRPs *PR-1*, and *PR-13 sulfur-rich thionin-like*. It also concerns some strain-specific modulations, with *PR-5* (which encodes the antimicrobial thaumatin-like protein), and *sesquiterpene synthase* (*SQTS*, which encodes the key enzyme of the mevalonate pathway involved in the synthesis of antimicrobial sesquiterpenes). It is highly plausible that the activity of these substances interferes with an optimal development of endophytes on wheat roots. The potential tolerance of root endophytes after their recognition by plants depends, *inter alia*, on their ability to suppress and/or detoxify specific plant defenses, and to protect their own vital cellular functions, the same ones that would be directed against pathogenic agents. The elicitation of these defenses is generally governed by SA- and JA-dependent signaling pathways, both of which appear to be elicited by the root endophytes isolated in our study. The fact that the negative modulation of these defenses is desynchronized from that of phytohormonal stress networks implies that beneficial microbes need to suppress specific local immune responses in the host, and that the regulation of these genes remains under the full control of PGPRs [191]. These singular transcriptomic data, albeit counterintuitive in a context of potentiated immune performance, echo the fact that specific defense-encoding PR proteins or those involved in the defensive secondary metabolisms can be modulated lower in resistant cultivars than in those prone to pathogen infection, and that overexpression/silencing of certain pathogenesis-related genes impairs or enhances resistance to various diseases [192–194]. The fact that certain defense genes are transcribed does not

systematically result in an accumulation of related products. For example, a common response of plants to fungal infection is the elicitation of GSL22, which by polymerizing the units of (1,3)-*β*-glucan together, allows the accumulation of callose at the site of infection by the pathogen, a form of cell wall thickening called papillae. The fact that the *GSL22* gene is significantly activated in leaves does not reflect a generalized cell wall strengthening of this organ, but a probable set-up of a durable and systemic transcriptomic defense arsenal that would be deployed more rapidly and more strongly in a situation of opportunist infections. This is reminiscent of the priming phenomenon [195] known to be elicited by beneficial organisms [196–198]. Priming is a durable physiological response, and it maintains, throughout the plant life cycle, an efficient immunity status against various stressors at no associated developmental cost to the plant, as was observed in our biological conditions with rhizobacteria (PGPR)-treated plants that exhibit similar or higher root and aerial growths as controls (Fig 2). In this light, the valorization of several new PGPR endophytes isolated in this study as priming plant defense inducers to combat biotic and abiotic stresses will contribute to promising and sustainable new eco-friendly strategies to improve better protection and management of field crops [199, 200].

## Conclusion

In this study, 14 bacterial endophytes isolated from healthy wheat roots significantly improved shoot and root biomass accumulation in wheat seedlings and triggered the plant's innate immune responses by stimulating the expression of various defense-encoding genes under greenhouse conditions. *In vitro*, the isolates showed several plant growth-promoting traits, including the production of indole-related compounds, ammonia, and HCN, and the solubilization of phosphate and zinc. The strains also showed positive results for the production of extracellular enzymes, such as catalase, amylase, protease, pectinase and cellulase, which are important for mitigating stress. To control *F. culmorum*, three isolates, identified as *Pantoea agglomerans*, *Paenibacillus peoriae*, and *Paenibacillus polymyxa*, demonstrated strong antagonistic properties *in vitro*. They exhibited a remarkable capacity to inhibit mycelium growth, sporulation, macroconidia germination, and germ tube length of *F. culmorum*, suggesting potential use as an alternative to chemical treatments while promoting plant growth. All these exploratory results are promising, but further experimental work is now needed to gain a better understanding of the efficacy of these strains under field conditions.

## Supporting information

**S1 Fig. Macroscopic aspect of the 14 isolated bacterial endophytes growing on agar-solidified lysogeny broth (LB) medium after 48 h of incubation at 37˚C.**
(PDF)

**S2 Fig. Macroscopic assessment and quantification of all biochemical activities studied for each strain.**
(PDF)

**S3 Fig. Macroscopic assessment of the dual confrontations of each rhizobacteria against *F. culmorum*, and semi-quantification of all CDWEs activities studied for each strain.**
(PDF)

**S1 Table. Primers used for the qPCR amplifications.**
(PDF)

**S2 Table. Statistical analysis of the transcriptomic data.**
(PDF)

## Acknowledgments

The authors thank those contributors who make the bacterial genome data accessible in public databases of NCBI. We are grateful to Amélie COSTON, Céline SAC, Caroline SAVEL for their technical assistance in handling of plants in greenhouse and in molecular biology, and Norbert FRIZOT for his technical computer support services. The authors wish to acknowledge Richard RYAN for providing the final linguistic revision of the manuscript. We are grateful to the Mésocentre Clermont-Auvergne and the plateforme AuBi of the Université Clermont Auvergne for providing help, computing and storage resources. Computations have been performed on the supercomputer facilities of the Mésocentre Clermont-Auvergne of the Université Clermont Auvergne. We are indebted to the anonymous reviewers for their helpful suggestions.

## Author Contributions

**Conceptualization:** Mouadh Saadaoui, Jean-Stéphane Venisse.

**Data curation:** Mohamed Faize, Patricia Roeckel-Drevet, Jean-Stéphane Venisse.

**Formal analysis:** Mohamed Faize, Patricia Roeckel-Drevet, Jean-Stéphane Venisse.

**Funding acquisition:** Jean-Stéphane Venisse.

**Investigation:** Jean-Stéphane Venisse.

**Methodology:** Mouadh Saadaoui, Mohamed Faize, Jean-Stéphane Venisse.

**Project administration:** Jean-Stéphane Venisse.

**Resources:** Jean-Stéphane Venisse.

**Software:** Patricia Roeckel-Drevet, Jean-Stéphane Venisse.

**Supervision:** Jean-Stéphane Venisse.

**Validation:** Mohamed Faize, Aicha Rifai, Patricia Roeckel-Drevet, Jean-Stéphane Venisse.

**Visualization:** Koussa Tayeb, Hatem Chaar.

**Writing – original draft:** Mouadh Saadaoui, Aicha Rifai, Koussa Tayeb, Noura Omri Ben Youssef, Mohamed Kharrat, Patricia Roeckel-Drevet, Jean-Stéphane Venisse.

**Writing – review & editing:** Mohamed Faize, Hatem Chaar, Jean-Stéphane Venisse.

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
