## [Decision Letter · Decision Letter 0]

8 Jan 2024

PONE-D-23-35387Evaluation of tunisian wheat endophytes as plant growth promoting bacteria and biological control agents against Fusarium culmorumPLOS ONE

Dear Dr. Venisse,

Thank you for submitting your manuscript to PLOS ONE. After careful consideration, we feel that it has merit but does not fully meet PLOS ONE’s publication criteria as it currently stands. Therefore, we invite you to submit a revised version of the manuscript that addresses the points raised during the review process.

We look forward to receiving your revised manuscript.

Kind regards,

Estibaliz Sansinenea

Academic Editor

PLOS ONE

“This work was awarded by the I-SITE CAP 20-25 (ANR grant 16-IDEX-0001) Emergence 2017 from the University of Clermont-Auvergne, and the project "Pack Ambition International 2019" [ANR grant n° P010O003] co-financed by the University of Clermont-Auvergne and the French Region "Auvergne-Rhônes-Alpes".”

Additional Editor Comments:

The reviewers have commmented about this MS. One of them has suggested minor revision however the second has rejected the MS due to is lacking of novelty.I have read the MS and although the topic has been widely stidied this Ms is experimentally sound and is well written. Therefore I invite the authors to do all revisio s suggested by the reviewers answering in the best way the second referee.

Reviewers' comments:

Reviewer's Responses to Questions

**Comments to the Author**

1. Is the manuscript technically sound, and do the data support the conclusions?

Reviewer #1: Yes

Reviewer #2: Partly

2. Has the statistical analysis been performed appropriately and rigorously? 

Reviewer #1: Yes

Reviewer #2: Yes

3. Have the authors made all data underlying the findings in their manuscript fully available?

Reviewer #1: Yes

Reviewer #2: Yes

4. Is the manuscript presented in an intelligible fashion and written in standard English?

Reviewer #1: Yes

Reviewer #2: Yes

5. Review Comments to the Author

Reviewer #1: Dear Corresponding Author

I checked your paper. It is well-written and in my idea everything is going excellent. I have some minor comments to improve your paper more:

1) In Figure 4 part B please use heat-map instead of +/- table.

2) About statistical analysis are you sure your data had normal distribution? We will use ANOVA just for normal data.

Regards

Reviewer #2: The research paper titled “Evaluation of tunisian wheat endophytes as plant growth promoting bacteria and biological control agents against Fusarium culmorum” was thoroughly examined.

Regrettably, it represents a competent yet repetitive investigation of endophytes as PGPR agents.

Numerous similar articles are readily accessible in databases such as Google Scholar, so the rationale behind the authors’ decision to replicate these types of experiments is not readily comprehensible to me.

It is inherently apparent from the outset that endophytes exhibit favorable effects on pathogenic agents.

6. PLOS authors have the option to publish the peer review history of their article (what does this mean?). If published, this will include your full peer review and any attached files.

Reviewer #1: No

Reviewer #2: **Yes: **Reza Khakvar

---

## [Author Response · Author response to Decision Letter 0]

4 Mar 2024

Dear Editor and Reviewers,

Thank you for your encouraging remarks on our work.

Please find below our answers for each Reviewers comments, which are also included in the "Reponse to reviewers" file. We have included our answer directly on the “View Letter” file by efficiency. All our replies are in red, as within the main text of the draft.

Concerning the Journal requirements:

-The full article was double-checked and we made every effort to ensure that our manuscript met PLOS ONE's stylistic requirements

-The mention "The funders had no role in study design, data collection and analysis, decision to publish, or preparation of the manuscript." was added in the “Funding” item in the article.

- We have removed the sentence which included the mention "Data not shown" from the discussion

Reviewer #1:

- We revised the Figure 4, making it easier to read as rightly suggested by reviewer 1. We added a color code to the "+" and "-" symbols.

- We confirm that our statistic analyses, normality tests were carried out for each series of values. we specify that : concerning ANOVA, normality of the data (residuals) was assessed by both visual inspection (Q-Q plot : quantile- quantile plot plot) and significance test (Shapiro-Wilk's test). In most cases, normality of the residuals was respected. In case the residuals were not normal, Tukey's Ladder of power transformation was used. This approach uses a simple power transformation on the data set (square root transformation, cube root transformation, etc.). Another thing, homogeneity (homoscedasticity) of variances test across groups also has to be carried out (Levene's test). In this respect, we have added a clarification in the statistical paragraph of the MM (lines 357-359).

Reviewer #2: 

We have no corrections to make, just a pertinent comment to which we are responding: "Thank you for emphasizing the rigor and the competence of our study. We are aware of the repetitive aspect of our study at first sight. However, it’s important to note that each pathosystem (as in this case Wheat - F. culmorum) possesses its own intrinsic specificity. Furthermore, the isolated strains exhibit their own singularity, which, in most cases, can be different between strains belonging to the same phyla. This variability is particularly pronounced when they evolve in a tripartite pathological context, which remains relatively complex.

You are right to point out that many similar articles are easily accessible in different databases. However, we believe it’s relevant to focus on isolating, testing and publishing the PGP and BCA capacities of the new bacterial strains isolated within a specific biotope. This approach is justified by the diverse effects that different PGPR strains can have on plants and their ability to protect against specific pathogens.

Furthermore, it's important to mention that this work constitutes a chapter in M. SAADAOUI's thesis. As such, this article represents a significant aspect of his academic development, and we are immensely proud of the positive evaluation it has received. We sincerely appreciate your recognition and encouragement."

We hope that we lived up to your expectations. Please do not hesitate to contact us for any additional information you may require.

Thank you for your comments and your help to improve our article. 

Sincerely yours, 

Dr Jean-Stéphane VENISSE

---

## [Editor Report · Decision Letter 1]

6 Mar 2024

Evaluation of tunisian wheat endophytes as plant growth promoting bacteria and biological control agents against Fusarium culmorum

PONE-D-23-35387R1

Dear Dr. Venisse,

We’re pleased to inform you that your manuscript has been judged scientifically suitable for publication and will be formally accepted for publication once it meets all outstanding technical requirements.

Kind regards,

Estibaliz Sansinenea

Academic Editor

PLOS ONE

Additional Editor Comments (optional):

The authors have done all corrections, tehrefore the MS can be accepted in the current form